# A machine learning model reveals expansive downregulation of ligand-receptor interactions that enhance lymphocyte infiltration in melanoma with developed resistance to immune checkpoint blockade

Sahil Sahni [1], Binbin Wang[1], Di Wu[2], Saugato Rahman Dhruba [1], Matthew Nagy[1], Sushant Patkar [3], Ingrid Ferreira [4], Chi-Ping Day[1], Kun Wang [1,5,6] ✉ & Eytan Ruppin [1,6] ✉

Immune checkpoint blockade (ICB) is a promising cancer therapy; however, resistance frequently develops. To explore ICB resistance mechanisms, we develop **I**mmunotherapy **R**esistance cell-cell **I**nteraction **S**canner (IRIS), a machine learning model aimed at identifying cell-type-specific tumor microenvironment ligand-receptor interactions relevant to ICB resistance. Applying IRIS to deconvolved transcriptomics data of the five largest melanoma ICB cohorts, we identify specific downregulated interactions, termed *resistance downregulated interactions* (RDI), as tumors develop resistance. These RDIs often involve chemokine signaling and offer a stronger predictive signal for ICB response compared to upregulated interactions or the state-of-the-art published transcriptomics biomarkers. Validation across multiple independent melanoma patient cohorts and modalities confirms that RDI activity is associated with CD8 + T cell infiltration and highly manifested in hot/brisk tumors. This study presents a strongly predictive ICB response biomarker, highlighting the key role of downregulating chemotaxis-associated ligand-receptor interactions in inhibiting lymphocyte infiltration in resistant tumors.

Immune checkpoint blockade (ICB) therapy provides durable clinical benefits to melanoma patients, but the response rate still has room for improvement, with approximately 40% of patients experiencing positive outcomes[1]. The majority of patients receiving checkpoint therapy face the challenge of developing resistance, including both primary resistance where the tumor does not respond to treatment initially and acquired resistance where initially responding tumors turn resistant over time[2–4]. The mechanisms of ICB resistance are complex and still not well understood. Current hypothesized mechanisms of resistance include depletion of neoantigen expression by tumor cells,

[1]Cancer Data Science Laboratory (CDSL), Center for Cancer Research (CCR), National Cancer Institute (NCI), National Institutes of Health (NIH), Bethesda, MD, USA. [2]Laboratory of Pathology, Center for Cancer Research (CCR), National Cancer Institute (NCI), National Institutes of Health (NIH), Bethesda, MD, USA. [3]Artificial Intelligence Resource, Molecular Imaging Branch, National Cancer Institute (NCI), National Institutes of Health (NIH), Bethesda, MD, USA. [4]Experimental Cancer Genetics, Wellcome Sanger Institute, Wellcome Genome Campus, Hinxton, Cambridge, UK. [5]Department of Comparative Biosciences, University of Illinois Urbana-Champaign, Urbana, IL, USA. [6]These authors jointly supervised this work: Kun Wang, Eytan Ruppin. ✉e-mail: kwang222@illinois.edu; eytan.ruppin@nih.gov

deficiencies in antigen-presentation machinery, and premature exhaustion of effector T cells[5]. Moreover, the composition of the tumor microenvironment (TME) can alter ICB response by composing an immunosuppressive or immunoreactive TME[5–7]. Identifying resistance mechanisms is crucial for developing combination therapies that target multiple resistance mediators simultaneously, improving patient response to ICB[5].

Numerous transcriptomic-based biomarkers have emerged for predicting ICB therapy response and uncovering mechanisms of resistance. *TIDE* infers gene signatures associated with T cell dysfunction and exclusion from TCGA using bulk transcriptomics[8]. *IMPRES* learned the pairwise relations of 15 checkpoint genes relevant to spontaneous regression in neuroblastoma using bulk transcriptomics[9]. *MPS* uncovered a melanocytic plasticity signature associated with ICB therapy resistance, derived from a combination of mouse models and bulk transcriptomics of ICB patients[10]. *Cytotoxic signature* identified a gene-expression signature correlated with aneuploidy and negative correlated with immune infiltration for melanoma patients using bulk transcriptomics[11]. *resF* elucidated a transcriptomic program associated with T cell exclusion and immune evasion from scRNA-seq of ICB-treated melanoma patients[12]. *Tres* identifies signatures of T cells resilient to immunosuppressive TME and confer antitumor properties from single-cell transcriptomics[13]. However, these biomarkers have notable limitations: 1. Bulk transcriptomic biomarkers overlook cell type heterogeneity within TME; 2. Single-cell transcriptomic signatures may not apply to the much more abundant bulk ICB cohorts; 3. Most of the pertaining genes composing these biomarker signatures are not targetable; and 4. Their predictive power leaves room for further improvement.

Cell-cell interactions in the TME plays a crucial role in impacting tumor growth and clinical outcome[14]. As a fundamental signaling mechanism, it is mediated by cell-type-specific ligand-receptor interactions, which ultimately underlie ICB therapy[5,6,14]. Characterization of ICB therapy resistance mediating interactions is likely to advance our understanding of the development of resistance, enhance ICB response prediction, and help discover novel therapeutic targets. However, established approaches for profiling cell-type-specific gene expression (e.g., FACS) do not directly provide such interaction data[15]. To dissect bulk transcriptomes and prioritize literature-curated, clinically relevant cell-cell interactions, we recently developed **CO**nfident **DE**convolution **F**or **A**ll **C**ell **S**ubsets (CODEFACS) and **LI**gand-**R**eceptor **I**nteractions between **C**ell **S**ubsets (LIRICS), respectively[15]. These methods have laid the basis for the development here of **I**mmunotherapy **R**esistance cell-cell **I**nteraction **S**canner (IRIS), a supervised machine learning method for identifying cell-type-specific ligand-receptor interactions relevant to ICB therapy response in the TME.

In this work, IRIS aims to discern a catalog of cell-type-specific ligand-receptor interactions pivotal in fostering resistance to ICB therapy, leveraging data from the most influential ICB therapy cohorts in melanoma[9,16–19]. Our findings underscore that these identified interactions hold promise as potent biomarkers for predicting ICB therapy response, surpassing previously published transcriptomic biomarkers. Moreover, functional analysis of these interactions offers profound insights into the underlying mechanisms of resistance development. Subsequently, we validate the potential functions of these interactions through comprehensive analysis of multi-modal transcriptomics datasets. In essence, our study introduces a machine learning approach for systematically identifying treatment resistance-relevant ligand-receptor interactions. Importantly, this versatile method holds potential for applications across diverse scenarios.

## Results

### Overview of IRIS and the analysis

We obtained three deconvolved melanoma ICB cohorts[16–18] from the CODEFACS publication and further deconvolved two additional melanoma ICB cohorts[9,19] using CODEFACS, a deconvolution method recently developed by our laboratory[15]. The expression profile of ten cell types in the TME (B cells, CD8 + T cells, CD4 + T cells, cancer-associated fibroblasts, endothelial cells, macrophages, malignant cells, natural killer cells, plasmacytoid dendritic cells, and skin dendritic cells) are derived for each patient's tumor sample from the ICB cohorts. Subsequently, we employed LIRICS[14], a ligand-receptor interaction inference tool, to derive cell-type-specific ligand-receptor interaction activity profile in each patient given the output from CODEFACS. The corresponding clinical information including response labels, survival and timelines were available for each patient.

We developed **I**mmunotherapy **R**esistance cell-cell **I**nteraction **S**canner (IRIS), a computational method specifically designed to identify immune checkpoint blockade (ICB) resistance relevant ligand-receptor interactions in the tumor microenvironment (TME), given a patients cohort including tumor bulk expression data and ICB treatment response data. The gene expression data is deconvolved using CODEFACS such that the input to IRIS in a given patients cohort is comprised of two components (Fig. 1A–B): 1. Literature-curated cell-type-specific ligand-receptor interaction activity profiles (denoting either activation: 1 or inactivation: 0) in each tumor sample, which is inferred using LIRICS from the deconvolved expression—an interaction is considered as activated if the (deconvolved) expression of both its ligand and receptor genes is above their median expression values across the cohort samples, and inactivated otherwise; 2. The corresponding ICB response outcome for each patient.

Based on both the ligand-receptor interaction activity and response data, IRIS employs a two-step supervised machine learning method to identify ICB resistance-relevant interactions (Fig. 1B)[1]. In the first step, the model selects interactions that exhibit differential activation between pre-treatment and post-treatment *non-responder* (progressive/stable disease) patients. The primary objective of this step is to identify interactions whose activation state following exposure to ICB is associated with the development of resistance. These interactions are categorized as *resistant downregulated interactions (RDI)* or *resistant upregulated interactions (RUI)*, based on their differential activity state in the post-treatment vs. the pre-treatment samples; that is, RDIs are downregulated in post-treatment resistant patients and vice versa for RUIs[2]. In the second step, we employ a hill-climbing aggregative feature selection algorithm to select an optimal set of ligand-receptor interactions that maximizes the classification power in distinguishing responders and non-responders from the pre-treatment tumor transcriptomics. When multiple training cohorts are available, both steps are executed iteratively on each cohort, resulting in cohort-specific models—an optimal set of response predictive, cell-type-specific ligand-receptor interactions for each cohort, which are composed of resistance downregulated interactions (RDI) and resistance upregulated ones (RUI). Additionally, each patient's *resistant upregulated score (RUS)* and *resistant downregulated score (RDS)* is computed for each tumor sample. RUS represents the normalized count of activated RUIs, while RDS represents the normalized count of activated RDIs. Higher RUS indicates non-responsiveness, while higher RDS indicates higher responsiveness to ICB therapy—the more active-state RDIs present in a tumor in the pre-treatment stage, the more likely the patient is to respond to the ICB therapy. In this study, we classify both RECIST criteria progressive and stable disease as forms of ICB resistance or non-response.

### Downregulated interactions in resistant patients offer stronger predictive value for ICB therapy response compared to upregulated ones

Functional resistance-relevant interactions are anticipated to be predictive of ICB therapy response. To evaluate and quantify their response prediction power, we employed a "leave one cohort out" strategy to assess the performance of RUS and RDS scores in

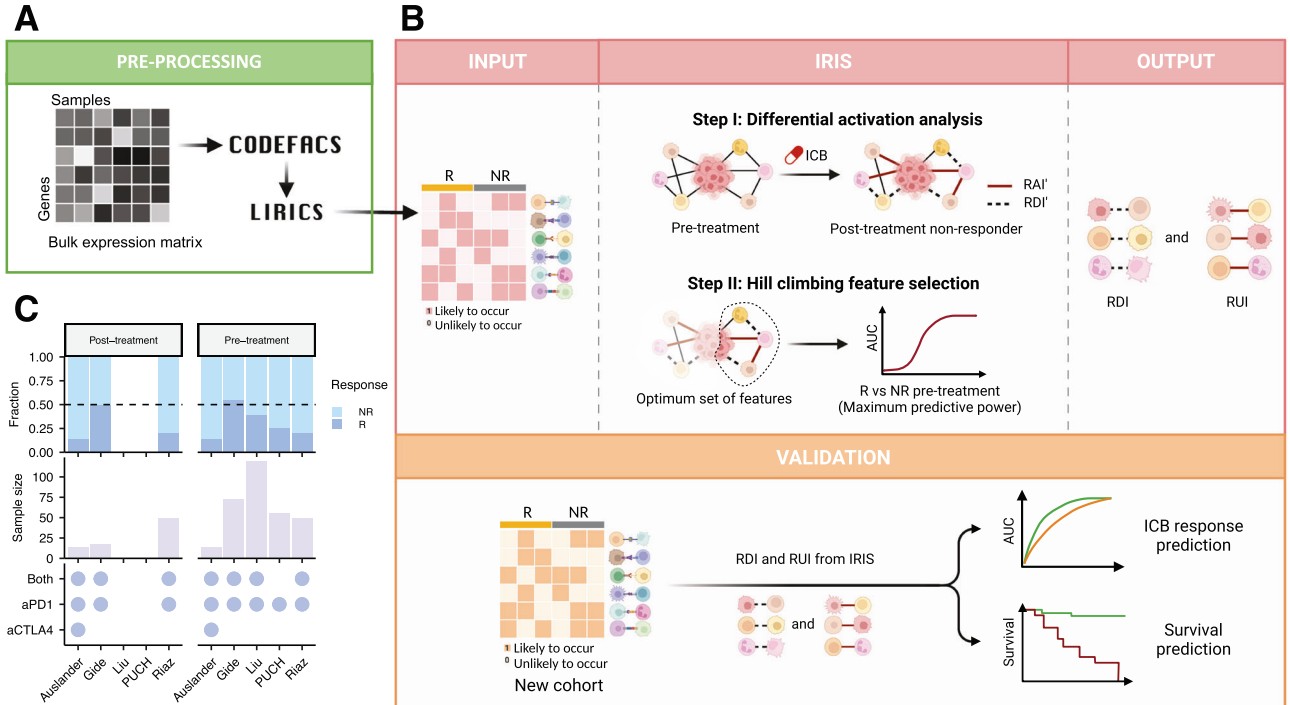

**Fig. 1 | Overview of IRIS. A–B** The IRIS input includes cell-type-specific ligand-receptor interaction activity profiles (inferred by applying CODEFACS and LIRICS on the tumor transcriptomics) and treatment response labels: responder (R) and non-responder (NR). It consists of two steps: Step I uses a Fisher's test to identify differentially activated ligand-receptor interactions in the pre-treatment and non-responder post-treatment samples. These interactions are categorized as either resistant downregulated interactions (RDI) or resistant upregulated interactions (RUI) based on their differential activity state in the post-treatment vs. the pre-treatment state; that is, RDIs are downregulated in post-treatment resistant patients and vice versa for RUIs. Step II employs a hill climbing aggregative feature selection algorithm to choose the optimal set of RDIs or RUIs for classifying responders and non-responders in pre-treatment samples. The final output of IRIS is a selected set of RDIs and RUIs hypothesized to facilitate in ICB resistance, that can be used to predict ICB therapy response in a new ICB cohort. **C** Demographics of deconvolved melanoma ICB cohorts used to train and validate IRIS. The X-axis indicates the ICB cohort name. The right and lefts panels correspond to post-treatment and pre-treatment samples to ICB respectively. The top panel depicts the relative proportion of responders (R) and non-responders (NR) samples in each cohort. The middle panel denotes tumor sample size in each cohort. The bottom panel displays the ICB treatment regimen administered, indicated with a purple dot, within each cohort: anti-PD1 monotherapy (aPD1), anti-CTLA4 monotherapy (aCTLA4), and combination therapy (aPD1 + aCTLA4). Figure 1 panel (**B**) created with BioRender.com, released under a Creative Commons Attribution-NonCommercial-NoDerivs 4.0 International license https://creativecommons.org/licenses/by-nc-nd/4.0/deed.en.

predicting ICB response. This involves iteratively designating each of the five cohorts as the *independent test cohort* while utilizing the remaining ones as the training cohorts to identify a set of resistance-relevant interactions using IRIS. Furthermore, the identified RUIs and RDIs from all the training cohorts, in aggregate, are utilized to derive RUS and RDS scores for ICB response prediction in the left-out test cohort.

We observe that the RDS significantly outperforms RUS in predicting ICB therapy response (one-sided paired Wilcoxon test $P = 0.0039$; Fig. 2A). The mean area under the curve (AUC) over all 5 independent test cohorts for RDS is 0.72, while for RUS it is a dismal 0.39 (Supplementary Fig. 1A). Furthermore, we conducted a more in-depth assessment on the association between each individual inter-action and ICB response in pre-treatment samples and post-treatment samples separately. The results reveal that RDIs are significantly more enriched for interactions relevant to ICB response compared to RUIs across both pre-treatment and post-treatment samples (Fisher's exact test $P = 3.3 \times 10^{-44}$ and $3.2 \times 10^{-40}$; odds ratio = ~18.8 and ~31.1 respectively; see Supplementary Fig. 2A–D). Moreover, we also studied the association of each individual interaction with both overall and progression-free survival benefit in the combined set of pre-treatments ICB samples. Similarly, the results reveal that RDIs are significantly more enriched for interactions associated with ICB overall and progression-free survival compared to RUIs (Fisher's exact test $P = 4.09 \times 10^{-10}$ and $3.98 \times 10^{-6}$; odds ratio = ~4.78 and ~3.28; see

Supplementary Fig. 2E–H). These findings together highlight the potential functional importance of RDIs in mediating ICB resistance, in contrast to the RUIs that lack predictive power. Therefore, we focus on RDIs in our subsequent analyzes.

Further assessing the predictive power of RDS, we find that responders exhibit significantly higher RDS compared to non-responders across five cohorts (Fig. 2B). Additionally, we find that the predictive power of RDS (average AUC: 0.72) is superior or comparable to five state-of-art ICB response predictors that emphasize resistance: TIDE (average AUC: 0.70; one-sided paired Wilcoxon test $P = 0.27$), IMPRES (average AUC: 0.62; one-sided paired Wilcoxon test $P = 0.055$), MPS (average AUC: 0.47; one-sided paired Wilcoxon test $P = 0.0078$), Cytotoxic signature (average AUC: 0.72; one-sided paired Wilcoxon test $P = 0.34$), and resF (average AUC: 0.58; one-sided paired Wilcoxon test $P = 0.055$) (Fig. 2C). To assess robustness, we calculated the coefficient of variation (CV = $\frac{sd(AUCs)}{mean(AUCs)} * 100$), based on the performance (AUC values) of each method across datasets. Our method's CV is substantially smaller (CV = 13.0) compared to others (CV = 24.7 for IMPRES; 23.3 for TIDE, 24.4 for MPS; 20.9 for Cytotoxic Signature; and 30.3 for resF), demonstrating greater consistency. We next benchmarked our RDS against other established transcriptomic markers of ICB therapy and immune response, including *PD1*, *PDL1*, *CTLA4*, and T cell exhaustion. These signatures exhibited high variability in their predictive performance while RDS consistently maintained considerable predictive power (Supplementary Fig. 1B). From a

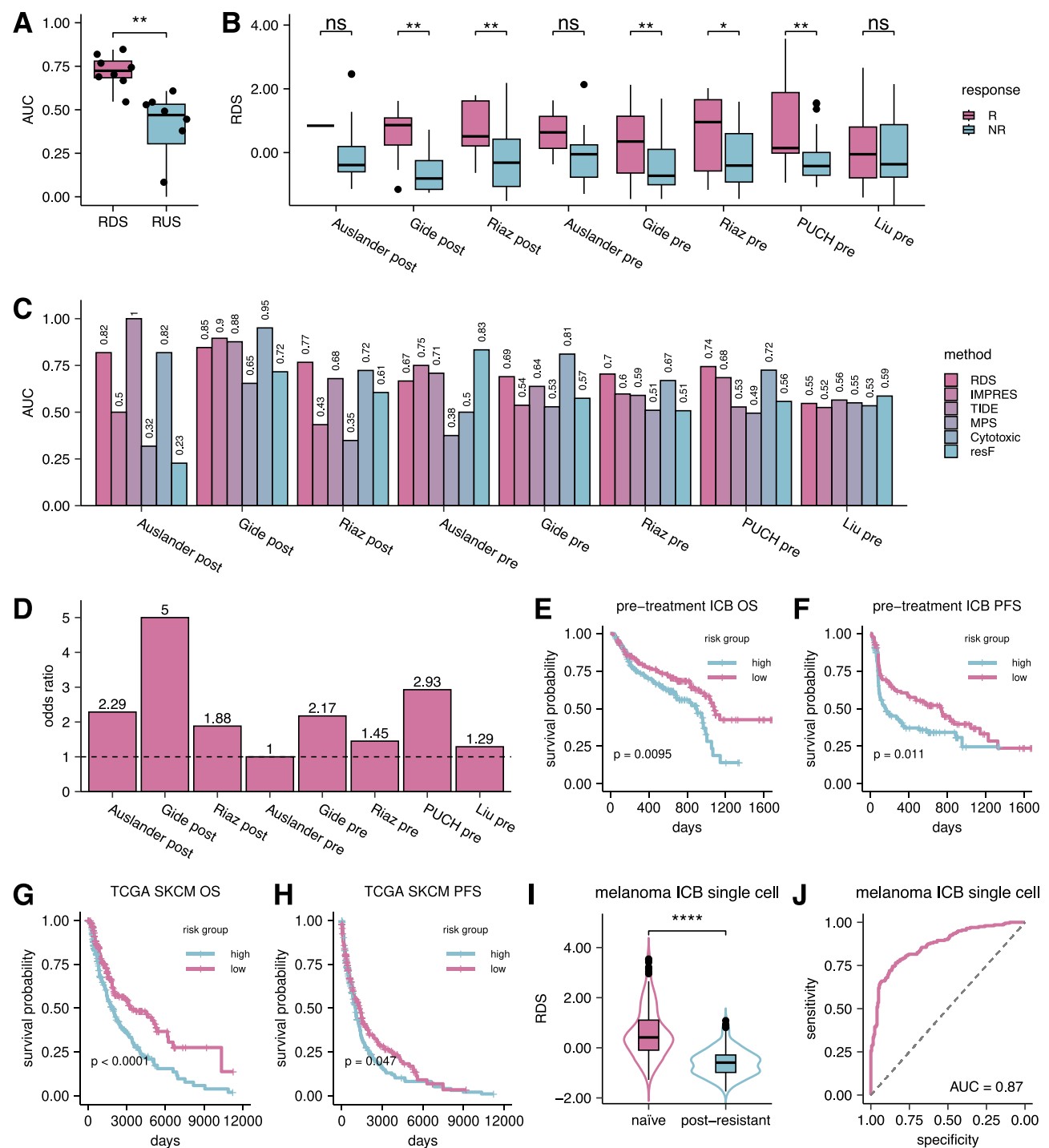

translational perspective, RDS accurately classifies responders and non-responders with a high odds ratio (average odds ratio = ~2.3; see Fig. 2D). More importantly, RDS significantly stratifies pre-treatment patients' survival outcomes with ICB therapy. Figure 2E shows the overall survival differences between low-risk (RDS score > median; $n = 146$) and high-risk groups (RDS score <= median; $n = 150$) across all ICB cohorts (log-rank test $P = 0.0095$). Similarly, Fig. 2F demonstrates significant stratification of low-risk ($n = 119$) and high-risk groups ($n = 122$) based on progression-free survival (log-rank test $P = 0.011$). IMPRES, TIDE, and other established ICB transcriptomic signatures failed to replicate these stratifications (Supplementary Fig. 3A-L), highlighting the functional relevance of our RDIs as survival and response biomarkers. Permuting the RDI activity profile (average AUC:

0.56) or patient treatment response (average AUC: 0.51) do not replicate these performances (Supplementary Fig. 1C).

Additionally, we assessed the performance of RDS in The Cancer Genome Atlas melanoma (TCGA-SKCM) cohort by calculating an RDS score for each tumor sample based on RDIs learned in the ICB cohorts. We find that RDS can effectively stratify low-risk and high-risk patients based on both overall survival (Fig. 2G) and progression-free survival (Fig. 2H), with log-rank test $p$-values of $8.4 \times 10^{-5}$ and 0.047, respectively. This suggests that RDIs reflect fundamental immune response mechanisms impacting patients' clinical outcome also in the absence of ICB therapy.

We confirmed the expression of the ligands and receptors from our inferred RDIs within the relevant cell types using single-cell

**Fig. 2 | RDS-based prediction of ICB response in melanoma. A** Boxplot depicting the distribution of AUCs in classifying responder vs. non-responder samples in all melanoma ICB cohorts ($n = 8$) between inferred resistant downregulated (RDIs/RDS) or resistant upregulated interactions (RUIs/RUS). One-sided paired Wilcoxon test $P = 0.0039$. **B** Boxplot depicting the distribution of resistance downregulated score (RDS) between responder (R) and non-responder (NR) melanoma ICB samples. One-sided Wilcoxon test $p$-values (from left to right): 0.22 ($n = 23$), 0.0076 ($n = 18$), 0.0051 ($n = 49$), 0.26 ($n = 14$), 0.0028 ($n = 73$), 0.025 ($n = 49$), 0.0035 ($n = 55$), and 0.20 ($n = 119$). **C** Bar plot depicting the AUC in classifying responder vs. non-responder melanoma samples for numerous published transcriptomic based prediction scores including RDS, IMPRES[9], TIDE[8], MPS[10], Cytotoxic signature[11], and resF[12]. **D** Bar plots depicting the odds-ratio in classifying true responder and non-responder samples using RDS scores. **E–F** Kaplan-Meier plot depicting overall (**E**) ($n_{high} = 150$, $n_{low} = 146$) and progression-free (**F**) ($n_{high} = 122$, $n_{low} = 119$) survival of the combined set of pre-treatment melanoma samples receiving immune checkpoint blockade. The patients are stratified into low-risk/high-risk groups based on the RDS median value. The significance of survival differences was estimated using the log-rank test. **G–H** Kaplan-Meier plot depicting overall (**G**) ($n_{high} = 224$, $n_{low} = 230$) and progression-free (**H**) ($n_{high} = 224$, $n_{low} = 231$) survival of the combined set of pre-treatment TCGA-SKCM samples. The patients are stratified into low-risk/high-risk groups as above. The significance of survival differences was estimated using the log-rank test. Log-rank test for TCGA-SKCM progression-free survival (**G**): $P = 8.4 \times 10^{-5}$. **I** Boxplot depicting the distribution of RDS between untreated (naïve) and post-checkpoint resistance (post-ICB resistant) samples from a melanoma ICB single-cell cohort ($n_{naïve} = 200$, $n_{resistant} = 200$). One-sided Wilcoxon test $P = 1.81 \times 10^{-37}$. **J** ROC curve depicting the classification accuracy of naïve vs. post-ICB resistant tumors in a melanoma ICB single-cell cohort ($n = 400$) based on the RDS scores of each tumor sample (Methods). The boxplots for figures **A–B** and (**I**) displays median, 25 and 75 percentiles (Q1 and Q3) as bounds of the box, and whiskers that extends from the box to a minima of Q1 - 1.5 × IQR and maxima Q3 + 1.5 × IQR (where IQR is interquartile range).

transcriptomics data from Jerby-Arnon et al.'s melanoma ICB study[12]. Employing a one-sample t-test for each gene within our RDI list, we evaluated expression levels within relevant cell types. Notably, across ten cell types, nine showed significant expression of over sixty percent of the inferred ligand and receptor genes (Supplementary Fig. 5). To study the role of RDIs in resistance development via single-cell transcriptomics, we further analyzed Jerby-Arnon et al.'s study by implementing both treatment response and timepoint labels[12]. Utilizing our in-house R tool, SOCIAL (**S**ingle-cell transcript**O**mics **C**ell-cell **I**nteraction **AL**gorithm; see Supplementary Fig. 11), which we've developed by integrating insights from Kumar et al.[20], Vento-Tormo et al.[21], our own LIRICS framework[15], we inferred activated ligand-receptor interactions from the single-cell transcriptomics data (see Methods). *Without any additional training*, we directly derived the RDS score for each tumor sample using RDIs inferred from bulk ICB cohorts (Methods). The RDS scores in treatment naïve samples are significantly (one-sided Wilcoxon test $P = 1.81 \times 10^{-37}$) higher than those in post-treatment non-responding samples (Fig. 2I). Moreover, the derived RDS scores classify naïve and post-ICB resistant tumors with an AUC of 0.87 (Fig. 2J). These results further show that RDIs are predictive of ICB resistance when assessed using single-cell transcriptomics.

### Functional analysis reveals the potential role of resistant downregulated interactions in mediating CD8 + T cell infiltration to the TME

To investigate the possible association between resistant downregulated interactions and the development of ICB resistance, we conducted a functional analysis on the union of interactions inferred across each ICB cohort specific model. Across cohort specific models, we find significant (one-sided Fisher's test $P = 6.05 \times 10^{-31}$) overlap of 122 RDIs (out of 299 total inferred RDIs, see Source Data for complete RDI network) in at-least two or more cohort models that we have studied (Fig. 3A), suggesting that IRIS inferences are robust despite the high levels of heterogeneity across patient samples. Notably, these 299 interactions within the RDI network (Source Data) originate from a diverse network of tumor-immune crosstalk involving all ten cell types in the TME. Through cell type enrichment analysis, we observe significant enrichment of ligand-expressing cell types including malignant cells (one-sided Fisher's test $P = 0.020$, odds ratio = 1.47) and natural killer (NK) cells (one-sided Fisher's test $P = 0.00014$, odds ratio = 1.95) (Fig. 3B). Enriched receptor-receiving cell types include macrophages (one-sided Fisher's test $P = 0.019$, odds ratio = 1.50), NK cells (one-sided Fisher's test $P = 0.04$, odds-ratio = 1.33), and plasmacytoid dendritic cells (pDC) (one-sided Fisher's test $P = 0.001$, odds-ratio = 2.29) (Fig. 3B). The high involvement of malignant cells suggests that the tumor actively regulates the production of ligands to evade immune response. Previous studies investigating chemokines associated with lymphocyte rich TMEs

in melanoma observed altered expression of *CCL2*, *CCL3*, *CCL4*, *CCL21*, *CXCL10*, *CXCL11*, and *CXCL13*, all of which are ligands expressed by the tumor within the RDI network (Source Data)[22,23]. Notably, tumor derived expression of CXCL10 has been hypothesized to localize effector CD8 + T cells via their CXCR3 receptor to the tumor site, an interaction observed within our RDI network that exhibits anti-tumor properties and is associated with immunotherapy response[24,25]. Additionally, the high presence of innate immunity pDC, macrophages, and NK cells testifies to their functional relevance in ICB resistance development. The innate immunity is known to play pivotal roles in the initiation of adaptive immunity, which results in downstream inflammation of the TME[26–28]. Within our RDI network, a key interaction identified is between the CXCL9 ligand and the CXCR3 receptor lying on dendritic cells and CD8 + T cells respectively (see Fig. 3D), which has been reported to enhance intratumorally CD8 + T cell response in the context of PD1 blockade in melanoma[29]. Furthermore, we observe an NK cell derived expression of XCL1 interacting with XCR1 receptor on dendritic cells, which has been associated with anti-PD1 response in melanoma[30]. Analysis of the enriched cell pairs further reveals the involvement of antigen-presenting cells, also showing that CD4 + T cells are actively interacting with NK cells and pDC cells (Supplementary Fig. 2G). Collectively, our results suggest that the crosstalk between the tumor and the immune system may be disrupted and downregulated as a mechanism for evading immune response that leads to resistance to ICB treatment.

Notably, reviewing the functional enrichment of the RDIs based on LIRICS curated annotations, there is a prominent over-representation of chemotaxis interactions (one-sided Fisher's test $P = 0.0008$, odds ratio = 3.21; Fig. 3C). These include interactions that have been previously shown to be associated with the recruitment and trafficking of effector CD8 + T cells to the tumor mass (see Fig. 3D) including CXCL9,10,11 expressed on tumors and cancer associated fibroblasts trafficking CD8 + T cells via their CXCR3 receptor[24,28,31]; CX3CL1 expressed within the TME recruiting CD8 + T cells via their CX3CR1 receptor[29]; CXCL12 expressed in tumors are involved in homing effector CD8 + T cells via their CXCR4 receptor[28,32]; CXCL16 ligand expressed by cancer associated fibroblasts cells attracting effector CD8 + T cells via their CXCR6 receptor[31]; and finally, CCL3,5 from NK and dendritic cells, which in turn activates CCR5 receptors on dendritic cells that recruit CD8 + T cells downstream[31].

Beyond examining the previous pertaining literature, we proceeded to further study the role of resistant downregulated interactions (RDIs) in recruiting CD8 + T cells by studying the association between the RDS scores and T cell infiltration levels in the TME of TCGA-SKCM samples (which has not been included in the training cohort inferring the RDS scores). Our analysis shows a strong positive correlation (Pearson R = 0.65, $P = 7.4 \times 10^{-57}$) between RDS score and

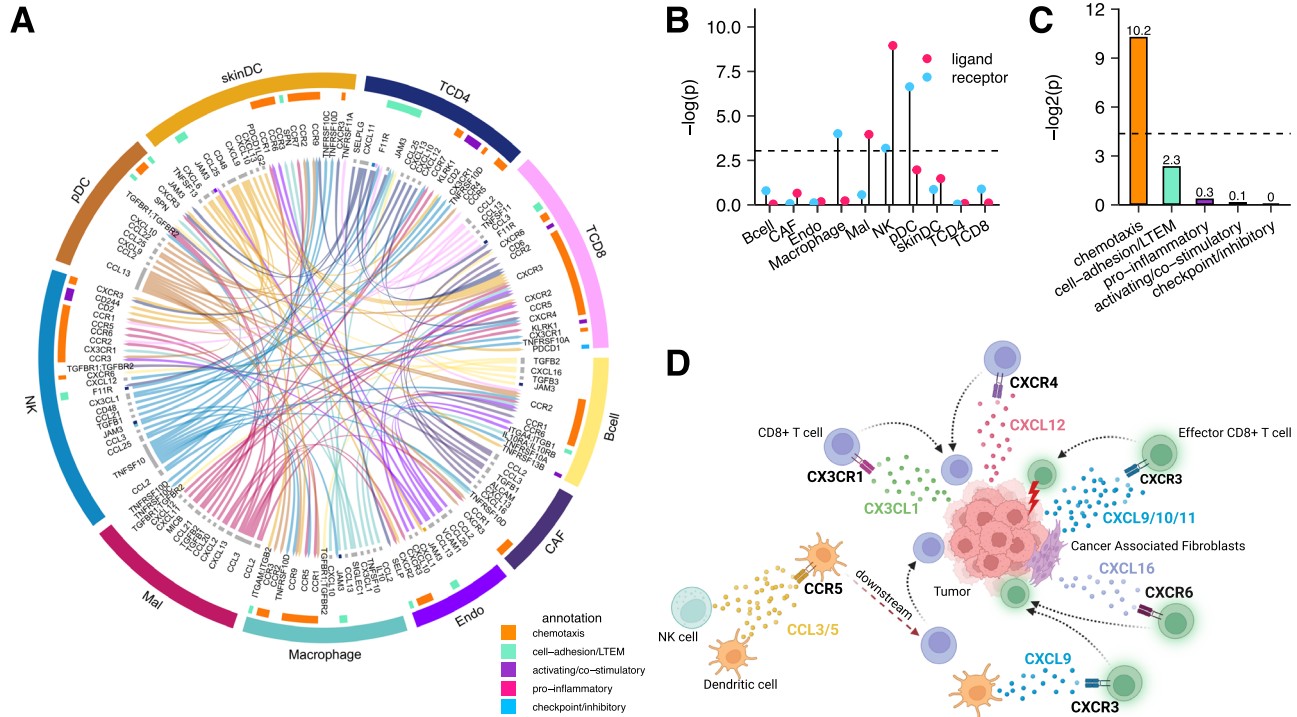

**Fig. 3 | Resistant downregulated interactions are enriched in those known to mediate CD8 + T cell infiltration to the TME. A** LIRICS' chord diagram representing the 122 interactions (out of 299 interactions in the RDI network) that are overlapped in at least two ICB cohort-specific models. Cell type abbreviations detailed in the methods. Literature-curated interaction functions from the LIRICS database are annotated in the inner ring and correspond to the colors detailed in the legend. Source data are provided as a Source Data file. **B** Lollipop plot depicting enrichment (one-sided Fisher's test) of individual cell types within RDI network as ligand-expressing (in red) or receptor-expressing (in blue) cells (the background is

all LIRICS' tumor-immune interactions; n = 3776). Cell types with enrichment p-values < 0.05 (dotted line) are shown. **C** Bar plot depicting enrichment (one-sided Fisher's test) of functional annotations within RDI network (background is the same as above). Functional annotations with p-values < 0.05 (dotted line) are shown. **D** Diagram displaying examples of cell-type-specific chemokine interactions found within the RDI network associated with CD8 + T cell infiltration in the TME supported by literature. Figure 3 panel (**D**), created with BioRender.com, released under a Creative Commons Attribution-NonCommercial-NoDerivs 4.0 International license https://creativecommons.org/licenses/by-nc-nd/4.0/deed.en.

---

the fraction of CD8 + T cells (inferred via CODEFACS, Methods) across TCGA-SKCM samples (Fig. 4A). Concomitantly, there is a strong negative correlation (Pearson R = −0.74, P = 9.5 × 10⁻⁸²) between RDS score and the fraction of malignant cells (Fig. 4B). We further examined the association between the RDS score and reported transcriptomic signatures of T cell infiltration (Fig. 4C), T cell exclusion (Fig. 4D), and post-resistance to ICB (Fig. 4E) in melanoma[12]. Notably, we find that RDS is significantly positively correlated (Pearson R = 0.78, P = 5.8 × 10⁻⁹⁶) with the T cell infiltration signature, while exhibiting negative correlations with the signatures of T cell exclusion (Pearson R = −0.73, P = 2.3 × 10⁻⁷⁸) and post-ICB resistance (Pearson R = −0.72, P = 3.8 × 10⁻⁸⁹). Signatures of exclusion and resistance were all controlled for T cell infiltration as recommended by Jerby-Arnon et al[12]. for more accurate prediction.

Additionally, we studied the association between RDS and T cell infiltration as quantified by expert pathologists' assessment of density and arrangement of infiltration in TCGA tumor slides. Tumors with broad intra-tumoral lymphocyte infiltration were termed as "brisk" patterns testifying to immunological "hot" TMEs, while tumors with partial or more focal lymphocyte infiltration are termed as "non-brisk" patterns and "cold" TMEs[33], where hot/brisk tumor niches have been associated with favorable prognosis in melanoma[33–35]. Based on these annotations we find that patients classified with hot tumor niches have significantly (one-sided Wilcoxon test P = 8.1 × 10⁻⁸) higher RDS scores than those with cold tumors (Fig. 4F; Supplementary Fig. 6E). Without any additional training, the RDS score classifies hot vs. cold tumors with an AUC of 0.66 in TCGA-SKCM (Supplementary Fig. 6F).

## RDIs are associated with CD8 + T cell fraction across spatial regions within the TME

To further elucidate the role of RDIs in modulating CD8 + T cell fraction across distinct regions of individual TMEs, we analyzed spatial transcriptomics data of metastatic melanoma samples sourced from both Thrane et al. and Biermann et al. studies[36,37]. Thrane et al. comprises of treatment naïve samples sequenced from the legacy platform, while Biermann et al. includes both treatment naïve and post-ICB resistant samples sequenced using the SlideSeqV2 platform. To quantify region specific activity of our inferred ligand-receptor interactions within each spatial transcriptomic slide, we extended our previously mentioned in-house R tool, SOCIAL to SPECIAL (**SP**atial transcriptomics c**E**ll-**C**ell **I**nteraction **AL**gorithm; Supplementary Fig. 12) for spatial transcriptomics by leveraging CytoSPACE[38]. In brief, we first aligned a reference single-cell and spatial transcriptomics data to deconvolve the mixtures within each region, achieving single-cell resolution using CytoSPACE. Utilizing a sliding window (or clustering) approach, we segmented the tumor slides into ~250 μm diameter regions, a distance conducive to a paracrine ligand-receptor interaction between individual cells[39]. Lastly, SOCIAL inferred the activity of individual cell-type-specific interactions within each spatial region (see Methods). We quantified the concordance between the estimated levels of RDI activity and CD8 + T cell fraction for each location on every slide. This involved utilizing the AUC metric after binarizing the CD8 + T cell fraction across regions, considering the sparsity of both quantities. Remarkably, we observe high levels of concordance (AUCs: median = 0.79; sd = 0.18 for Thrane et al. and median = 0.62; sd = 0.14 for Biermann et al.) between both measurements across cohorts and

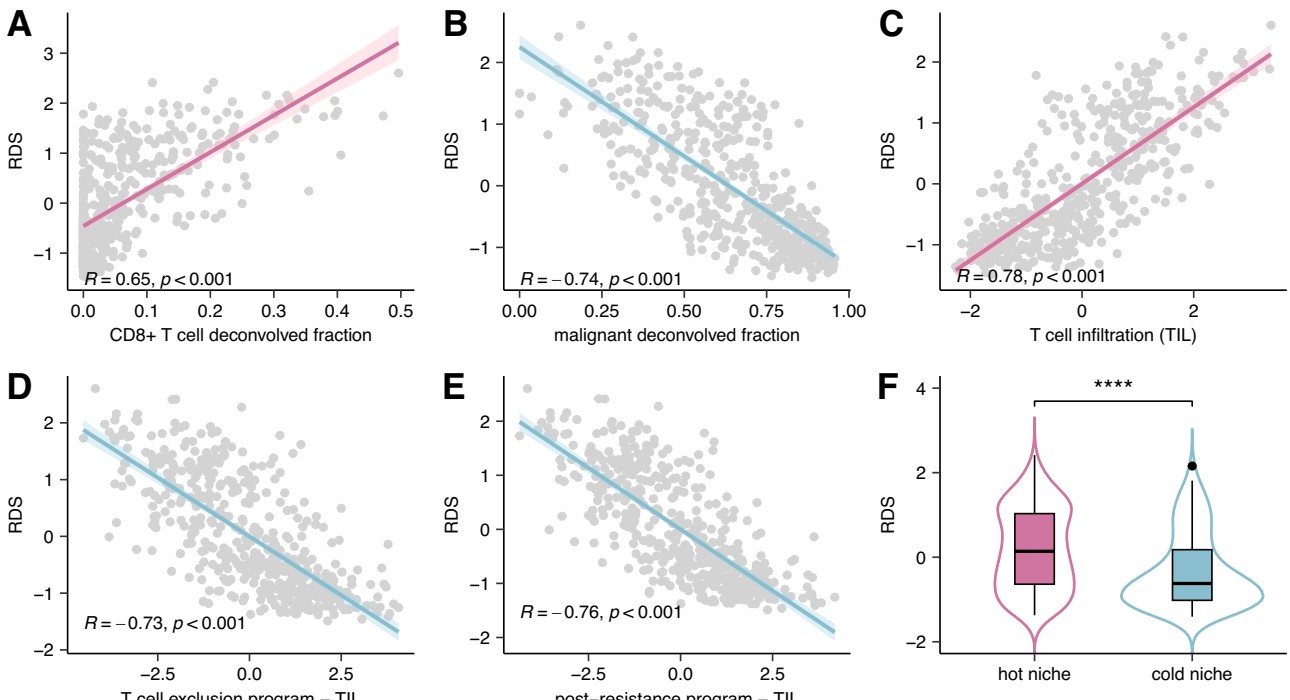

**Fig. 4 | RDS scores are associated with measures of T cells infiltration into the TME. A** A scatter plot depicting the correlation between RDS and estimated CD8 + T cell fraction from CODEFACS in TCGA-SKCM. Pearson R = 0.65, $P = 7.4 \times 10^{-57}$ ($n = 469$). **B** Scatter plot depicting the correlation between RDS and estimated tumor (malignant) fraction from CODEFACS in TCGA-SKCM. Pearson R = -0.74, $P = 9.5 \times 10^{-82}$ ($n = 469$). **C** Scatter plot depicting the correlation between RDS and transcriptomic signatures of T cell infiltration (TIL)[12] in TCGA-SKCM. Pearson R = 0.78, $P = 5.8 \times 10^{-96}$ ($n = 468$). **D** Scatter plot depicting the correlation between RDS and transcriptomic signatures of T cell exclusion[12] in TCGA-SKCM. The X-axis indicates the T cell exclusion program score adjusted for TIL. Pearson R = -0.73,

$P = 2.3 \times 10^{-78}$ ($n = 468$). **E** Scatter plot depicting the correlation between RDS and transcriptomic signatures of post-ICB resistance program[12] in TCGA-SKCM. The X-axis indicates the post-ICB resistance program adjusted for TIL. Pearson R = -0.72, $P = 3.8 \times 10^{-89}$ ($n = 468$). **F** Boxplot depicting distribution of RDS between hot (brisk) and cold (non-brisk) tumor niches in TCGA-SKCM ($n_{hot} = 223$, $n_{cold} = 146$). One-sided Wilcoxon test $P = 8.1 \times 10^{-8}$. The boxplot displays median, 25 and 75 percentiles (Q1 and Q3) as bounds of the box, and whiskers that extends from the box to a minima of Q1 - 1.5 × IQR and maxima Q3 + 1.5 × IQR (where IQR is inter-quartile range).

platforms (Fig. 5A, B; Supplementary Fig. 8A, B). Biopsy 3 (labeled as mel3_rep1 in Fig. 5A, B) likely underperforms due to the absence of any lymphoid tissue, as indicated by both manual and computational annotations in the original study[37]. Additionally, we quantitatively demonstrated that spatial regions with CD8 + T cell infiltration exhibited significantly (one-sided Wilcoxon test) more activated RDIs compared to regions lacking infiltration across both slide-specific (Supplementary Fig. 9A, B) and patient-specific resolutions (Fig. 5C, E) in both spatial cohorts. Furthermore, spatial regions from biopsied slides with CD8 + T cell infiltration ($n = 4$) harbored significantly (one-sided Wilcoxon test, $P = 3.83 \times 10^{-10}$; see Fig. 5D) more activated RDIs compared to spatial slides with CD8 + T cell deserts ($n = 4$) from Thrane et al.'s study. These findings collectively underscore the role of RDIs in enhancing CD8 + T cell infiltration when activated at a spatial resolution.

Furthermore, to investigate the role of RDIs in the development of ICB resistance using spatial transcriptomics, we initially compared treatment naïve and post-ICB resistant samples from Biermann et al.'s cohort. Our analysis revealed that spatial regions from resistant tumors had significantly (one-sided Wilcoxon test, $P = 0.0282$; see Fig. 5F) lower levels of activated RDIs compared to regions within treatment naïve tumor slides. The trend persisted at a per-patient level (one-sided Wilcoxon test, $P = 0.32$; see Fig. 5G), where resistant patients ($n = 2$) harbored fewer activated RDIs across all regions compared to treatment-naïve patients ($n = 6$). Additionally, we analyzed treatment naïve samples from Thrane et al. by scoring each region using six state-of-the-art ICB predictors directly from the spatial transcriptomics expression. Remarkably, we observed that spatial regions with RDI activity presented significantly higher ICB signature score,

indicative of ICB response, compared to regions without RDI activity at a per patient level (Supplementary Fig. 10). These findings collectively underscore both the downregulation of RDIs during the development of ICB resistance and the functional importance of their activity in predicting ICB response at a spatial resolution.

Collectively, our findings suggest that RDS is predictive of ICB therapy response and the identified RDIs are those involved in trafficking CD8 + T cells to the tumor site. We have proposed a theoretical mechanism underlying the development of resistance to ICB: Initially, those identified RDIs play a crucial role in recruiting CD8 + T cell to the tumor site and maintaining an active immune infiltrating environment sustainably. Pre-treatment activation of RDIs results in high CD8 + T cell infiltration levels in the TME reinforcing a hot tumor niche, albeit with a limited fraction of effector T cells due to checkpoint activation (Fig. 6 pre-treatment phase). Following ICB treatment, checkpoint interactions are blocked, leading to the transition of CD8 + T cells into a cytotoxic state, accompanied by a marked increase in effector T cell population. Consequently, the tumor size is reduced (Fig. 6 post-treatment phase). To evade immune attacks, resistant tumors deplete activated RDIs, disrupting the machinery trafficking lymphocytes to the TME via cell-cell commutation-mediated mechanisms. This reduces the lymphocyte infiltration, converting the previously "hot" TME to "cold" (Fig. 6 resistance phase). Consequently, there is a concomitant decrease in effector T cell fraction and an increase in tumor volume.

## Discussion
Identification of resistance factors relevant to cancer immunotherapy can enhance our understanding of resistance mechanisms and uncover novel therapeutic targets. Here, we present IRIS, a

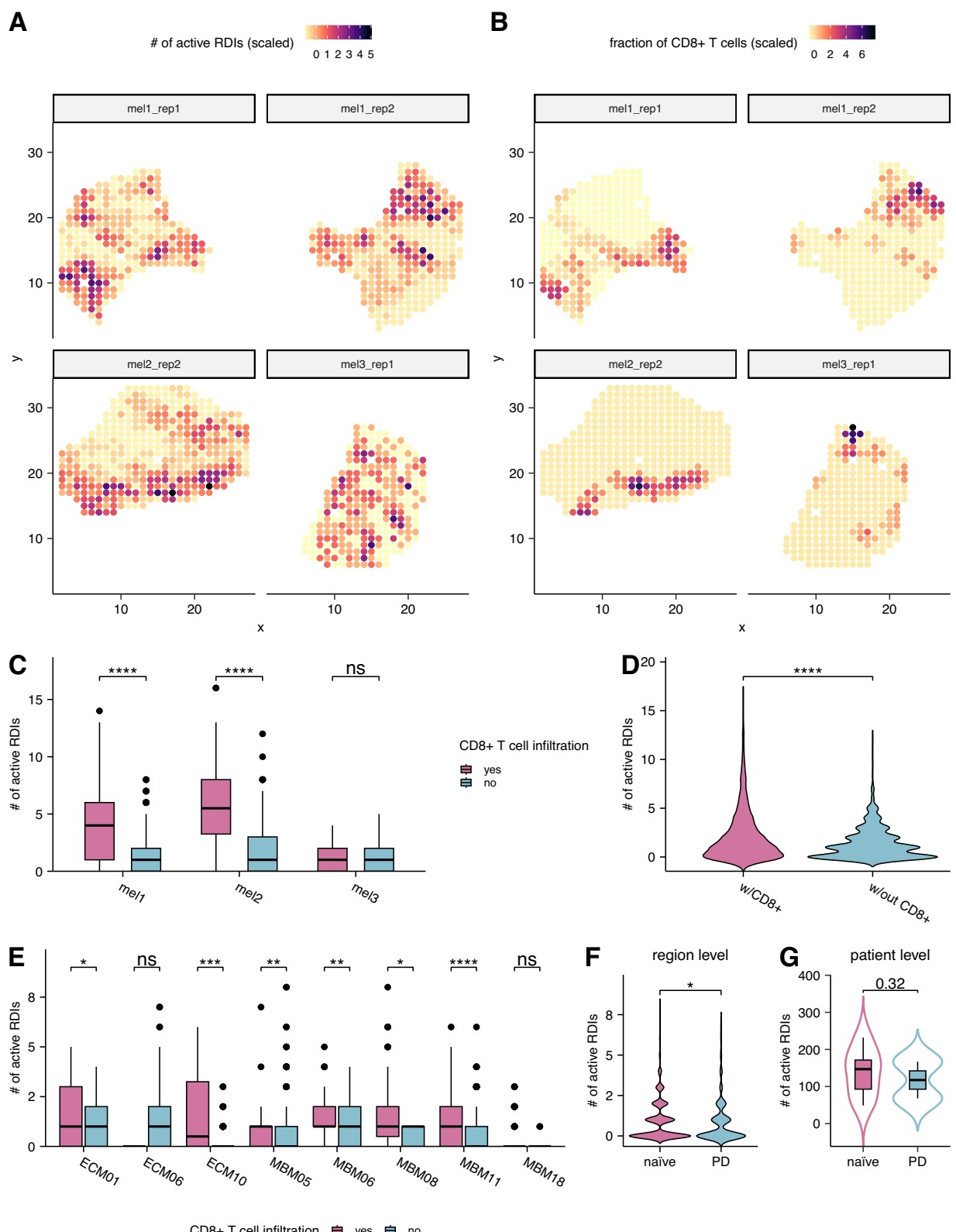

computational tool that identifies Immune Checkpoint Blockade (ICB) resistance associated ligand-receptor interactions in the tumor microenvironment (TME). Analyzing five large melanoma cohorts, we identify such predictive interactions, that when downregulated are associated with ICB resistance. An independent validation in a single-cell ICB cohort further supports these findings. These interactions are

highly enriched in stimulatory chemokine recruitment of CD8 + T cells to the tumor mass and are associated with "hot" tumor niches.

Our findings suggest that the inhibited recruitment of T cells to the tumor site is mainly associated with the loss of stimulatory chemokine signaling. Accordingly, following ICB treatment, the tumor actively modulates the TME to downregulate interactions that

**Fig. 5 | Association between RDIs activity and T cell infiltration measures within the TME at spatial resolution. A** Scatter plot illustrating the number of activated RDIs (scaled) within each spatial region of legacy spatial transcriptomics slides ($n = 4$). The *X*-axis corresponds to the x-coordinate of the spatial region, and the *Y*-axis corresponds to the y-coordinate of the spatial region. **B** Corresponding scatter plots depicting the fraction of CD8 + T cell infiltration (scaled) within each spatial region of the four legacy spatial transcriptomics slides. **C** Boxplot demonstrating the total number of activated RDIs between spatial regions with and without CD8 + T cell infiltration at a patient level, based on legacy spatial transcriptomics slides ($n = 4$). One-sided Wilcoxon test *p*-values (from left to right): $2.42 \times 10^{-30}$ ($n_{regions}=596$), $9.51 \times 10^{-20}$ ($n_{regions}=380$), and 0.73 ($n_{regions}=255$). **D** Violin plot illustrating the total number of activated RDIs within spatial regions from legacy slides with ($n = 4$) and without ($n = 4$) any CD8 + T cell infiltration. One-sided Wilcoxon test $P = 3.83 \times 10^{-10}$ ($n_{regions}=2336$). **E** Boxplot showcasing the total number of

activated RDIs between spatial regions with and without CD8 + T cell infiltration at a patient level, based on SlideSeqV2 spatial transcriptomics slides. One-sided Wilcoxon test *p*-values (from left to right): 0.046 ($n_{regions}=288$), 0.86 ($n_{regions}=144$), 0.00048 ($n_{regions}=143$), 0.0021 ($n_{regions}=408$), 0.0037 ($n_{regions}=144$), 0.011 ($n_{regions}=144$), $4.19 \times 10^{-8}$ ($n_{regions}=432$), and 0.38 ($n_{regions}=143$). **F** Violin plot displaying the number of activated RDIs across spatial regions between treatment-naïve and resistant/progressive disease (PD) patients ($n_{naïve}=2$, $n_{PD} = 6$) to ICB, based on SlideSeqV2 slides. One-sided Wilcoxon test $P = 0.0282$ ($n_{regions}=1846$). **G** Violin plot indicating the average number of activated RDIs between treatment-naïve and resistant/progressive disease (PD) patients ($n_{naïve}=2$, $n_{PD} = 6$) to ICB, based on SlideSeqV2 slides. The boxplots for figures (**C**) and (**E**) displays median, 25 and 75 percentiles (Q1 and Q3) as bounds of the box, and whiskers that extends from the box to a minima of Q1 - 1.5 × IQR and maxima Q3 + 1.5 × IQR (where IQR is interquartile range).

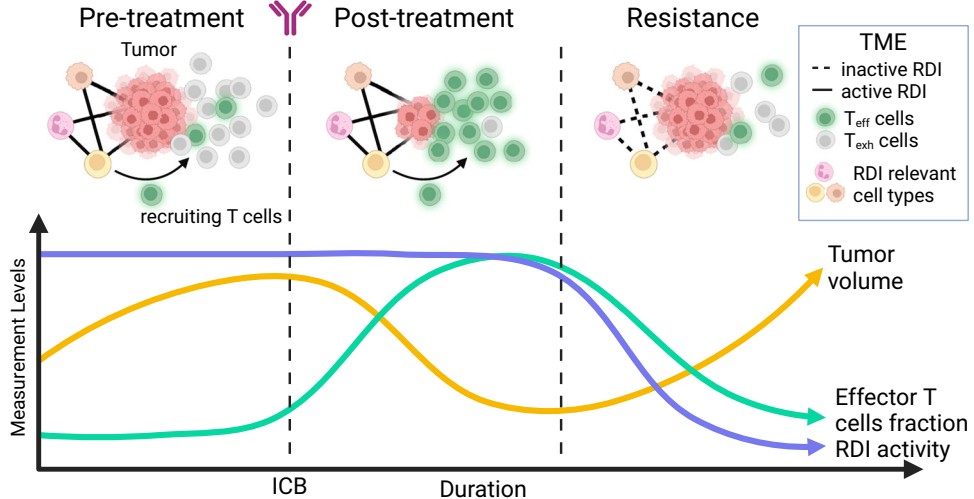

**Fig. 6 | Hypothesized evolution of immune microenvironment in melanoma during ICB treatment.** Based on our findings and existing knowledge, we propose a model of melanoma resistance following the initial response to ICB. In the pre-treatment stage, T cells are recruited to the tumor microenvironment by activated RDIs (purple curve, ligand-receptor pairs identified in this study) between various cell types (e.g., macrophages, DCs, etc.). However, the immune checkpoint interactions (e.g., PD1-PDL1) are preserved, reducing the ratio of effector T cells (green curve, referred to as $T_{eff}$ cells), allowing tumor cells to expand (yellow curve). In the

post-treatment stage, the ratio of effector T cells increases due to blocked checkpoint interactions, resulting in the initial response to reduce tumor volume. In the resistance stage, as a mechanism for tumors to overcome checkpoint blockade, RDI activities are declined resulting in decreased T cell recruitment. The progressive decline leads to diminish effector T cell population and therefore allowing tumor recurrence. Figure 6, created with BioRender.com, released under a Creative Commons Attribution-NonCommercial-NoDerivs 4.0 International license https://creativecommons.org/licenses/by-nc-nd/4.0/deed.en.

promote CD8 + T cells infiltration, as specifically described in the pertaining Results section[24,25,28,29,31,32]. Consequently, the TME transitions from a "hot" to a "cold" niche. These findings emphasize the importance of investigating stimulatory chemokine signaling in maintaining CD8 + T cell infiltration within the TME of ICB therapy responders[23,28,40,41]. Indeed, our findings are aligned with recent reports of interventions aimed at enhancing stimulatory chemokine expression to improve anti-tumor response, which are currently under investigation in pre-clinical models[41–47].

The RDS score can serve as a predictive biomarker for ICB response, complementing existing transcriptomic biomarkers. Importantly, RDS offers several advantages over previous biomarkers: 1. It can be effectively applied to bulk, single-cell, and spatial transcriptomics. 2. It exhibits robust predictive power across many different cohorts. 3. The identified interactions may uncover potential therapeutic targets, opening new avenues for therapeutic intervention. However, it is important to acknowledge the limitations of our study and identify areas for improvement. Firstly, like any predictive modeling approach, we identify associations and not causal mechanistic factors. One potential approach to address this is to incorporate

additional functional layers that can further refine the selection of interactions that are more likely to be drivers. For example, recent studies have proposed methods to explore the function of downstream transcription factors that are regulated by ligand-receptor interactions[48–50]. In our context, more causally related factors are expected to regulate specific transcription factors that play a crucial role in anti-tumor activity. Secondly, our study "*only*" considered well-defined ligand-receptor interactions from the existing literature instead of inferring de-novo cell-type-specific interactions. Furthermore, treatments administered fall into three distinct categories (anti-PD1, anti-CTLA4, and combination therapy with both anti-PD1 and anti-CTLA4 agents). Regrettably, due to the limited sample size within each treatment category of the training data, we were unable to extract treatment-specific insights. However, prospective studies in the future could delve into this direction provided there is access to more abundant samples. Thirdly, there is a lack of comprehensive functional annotations for ligand-receptor interactions. This is particularly relevant for chemokine interactions, where the same ligand-receptor pair can mediate both stimulatory and inhibitory effects in immune trafficking across different target cell types[24,31,51]. Moreover, varying

concentrations of chemokine ligands can either mediate or prevent immune infiltration in the TME. For example, CXCL12-CXCR4 axis facilitates recruitment of CD8 + T cells to the TME. However, at high CXCL12 concentrations the effects become inhibitory[28,32,52]. To address this knowledge gap, one potential solution is to computationally assess the association between each interaction and a clinical phenotype of interest across diverse cell types, concentrations, and contexts, which is of course a complex challenge on its own. Lastly, while our RDIs are composed of ligand-receptors situated on the surface of cells, targeting such large number of interactions in concert is infeasible. Hopefully, future prospective studies could aim to identify a subset of hotspot receptor targets from these interactions for potential therapeutic intervention[14].

In summary, our study presents an approach for identifying ICB resistance-relevant interactions using bulk deconvolved transcriptomics. Its application testifies that ICB resistance in melanoma is primarily associated with the downregulation of cell-type-specific interactions that regulate CD8 + T cell trafficking and infiltration within the tumor microenvironment.

## Methods

### Bulk RNA-seq datasets
We collected five publicly available bulk RNA-seq datasets from formalin-fixed-paraffin-embedded (FFPE) and fresh-frozen tumor tissue biopsies, accompanied by clinical information of melanoma patients undergoing immune checkpoint blockade (ICB) therapy. These datasets encompass Auslander et al. ($n = 37$), Gide et al. ($n = 91$), Riaz et al. ($n = 98$), Liu et al. ($n = 119$), and PUCH ($n = 55$) datasets[9,16–19]. The collection of datasets was frozen by Mar 2023. Each bulk RNA-seq dataset comprises a minimum of 30 samples, ensuring reliable deconvolution for ten distinct cell types. To ensure consistency for deconvolution using CODEFACS we normalized gene expression values to transcripts per million (TPM) for each dataset, following the guidance of Wang et al.[15]. Notably, all patients across these clinical studies received anti-PD1 monotherapy or a combination of anti-PD1 with anti-CTLA4 except Auslander et al.[9], which includes samples ($n = 6$) treated exclusively with anti-CTLA4 monotherapy. Across all cohorts, we binarized patients' ICB response to either responder (RECIST criteria: complete/partial response) or non-responder (RECIST criteria: progressive/stable disease). For Auslander et al., Gide et al., and Riaz et al., both pre- and post/on-treatment samples were collected. For Gide et al., Riaz et al., and Liu et al., TPM expression data, along with CODEFACS' deconvolved expression and cell fraction data for ten cell types were acquired from Wang et al.[15]. For Auslander et al., expression data was downloaded from GEO (GSE115821), while expression and clinical data for PUCH was downloaded from (https://github.com/xmuyulab/ims_gene_signature). In the PUCH dataset, we converted the overall survival duration into days by multiplying the reported months by 30.

In addition, we obtained the deconvolved expression and cell fraction data utilizing CODEFACS for skin cutaneous melanoma patients in the The Cancer Genome Atlas (TCGA-SKCM) from Wang et al.[15]. Overall survival and progression free interval (synonymous to progression free survival) for TCGA-SKCM patients was downloaded from the UCSC Xena browser[53] (https://xenabrowser.net). The pathology classification (brisk or non-brisk) of TCGA-SKCM samples was acquired from Saltz et al.[33]. Indeterminate ($n = 8$) and "none" ($n = 5$) pathology classifications were excluded from the analyses.

### Deconvolution of bulk ICB RNA-seq datasets using CODEFACS
To deconvolve newly acquired ICB datasets, we utilized cell-type-specific signatures for ten distinct cell types sourced from Wang et al.[15]. These signatures are tailored to melanoma tumors, encompassing key cell types that best represent the melanoma TME, comprising malignant cells (Mal), skin dendritic cells (skinDC), plasmacytoid dendritic

cells (pDC), CD8 + T lymphocytes (TCD8), CD4 + T lymphocytes (TCD4), macrophages, natural killer cells (NK), B lymphocytes (Bcell), endothelial cells (Endo), and cancer associated fibroblasts (CAF). Leveraging the provided signature and bulk datasets, CODEFACS[15] estimated the cell-type-specific expression and cell fractions for the identical set of ten cell types across all five deconvolved melanoma ICB cohorts.

### Inferring "activated" and "inactivated" cell-type-specific ligand-receptor interactions using LIRICS
Deconvolved expression data from the five melanoma ICB cohorts and TCGA-SKCM were inputted into LIRICS[15], which infers cell-type-specific ligand-receptor interactions, classified as "activated"[1] or "inactivated"(0) for each tumor sample within a given cohort. This procedure was conducted for pre-treatment samples and post-treatment samples separately. The scope of these inferred interactions is confined to those cell-type-specific ligand-receptor interactions present within the original curated LIRICS database ($n = 3776$).

### Inferring activated cell-type-specific ligand-receptor interaction from single-cell transcriptomics using SOCIAL
We developed an R code, SOCIAL (Single-cell transcriptOmics Cell-cell Interaction ALgorithm), to identify significant ligand-receptor interactions between two specific cell types, drawing upon insights from Kumar et al.'s[20], Vento-Tormo et al.'s[21], and our own LIRICS framework[15]. Our decision to create our own code stemmed from four primary motivations: 1. Leveraging the strengths of previous methods: By combining aspects of the three approaches, we aimed to maximize the accuracy and robustness of our ligand-receptor interaction predictions. 2. Implementing an R-based solution: While the first method lacked publicly accessible code and the second was in Python, we sought to create an R-based solution for accessibility and ease of use. 3. Incorporating our comprehensive database: Our ligand-receptor interaction database (LIRICS) provided rich and informative annotations, enhancing the depth of our analysis. 4. Accommodating variations in ligand-receptor interaction activity observed across patients.

SOCIAL (see Supplementary Fig. 11) comprises three main steps: 1. Querying the LIRICS database: Initially, we queried the LIRICS database to identify plausible ligand-receptor interactions; 2. Computing interaction scores: Next, we computed the ligand-receptor interaction score by multiplying the average expression levels of the ligand and receptor complexes for each interaction pair and cell type. 3. Permutation testing: Following that, we performed permutation tests (utilizing 100 iterations in our study) by randomly shuffling cell type labels. This allowed us to derive empirical $p$-values by calculating the fraction of permutation tests resulting in a higher interaction score than the foreground score determined in step 2. A lower $p$-value suggests a higher likelihood of the interaction occurring. 4. Optionally, ligand-receptor interactions can be further denoted as significantly activated if the average expression level of both the ligand and receptor genes is greater than the median across all samples.

### Validate resistant downregulated interactions using single-cell RNA-seq dataset
We obtained single-cell RNA-seq data (scRNA-seq) in TPM encompassing both naïve/untreated ($n = 16$) and post-ICB resistant ($n = 15$) resected melanoma tumors from Jerby-Arnon et al.[12]. Expression data was downloaded from GEO (GSE115978). Unfortunately, the post-ICB responder sample was omitted from our analyzes due to its limited patient sample size ($n = 1$). To ensure consistency across the bulk datasets and incorporate dendritic cells (DCs), which play a crucial role in immune response through antigen presentation modulation, we reannotated subsets of macrophages as DCs. Utilizing K-means clustering and PCA analysis on twelve classical DC marker genes[54,55], this expanded the number of cell types from eight to ten. The reannotation

encompassed skin resident DCs (a blend of dermal and epidermal DCs) along with plasmacytoid DCs. Additionally, T cells that did not align with CD4+ or CD8+ subtypes were excluded. Given the variances in available cell types across each tumor sample, we initially consolidated all cells within each treatment group (naïve or post-ICB resistant). Subsequently, we down sampled 40% of cells for each cell type to create "pseudo-samples" corresponding to their respective treatment group. This procedure was repeated through 200 iterations for each treatment group, yielding 200 naïve and 200 post-ICB resistant pseudo tumor samples. To discern the "activated" or "inactivated" status of curated LIRICS interactions using scRNA-seq data, we employed our proprietary R tool, SOCIAL, as discussed earlier, to identify activated ligand-receptor interactions. Interactions meeting the criteria of an empirical $p$-value < 0.05 and both ligand and receptor mean expression levels exceeding the median across all samples (similar to LIRICS methodology) were classified as "activated" through this procedure.

## Summary of IRIS

IRIS (Immunotherapy Resistance cell-cell Interaction Scanner) operates as a supervised machine learning algorithm, following a two-step process: 1. Differential activation analysis between pre-treatment and post-treatment non-responder group to extract a set of cell-type-specific ligand-receptor interactions, and 2. Employ a hill climbing aggregative feature selection algorithm to further optimize the selected interactions from step 1 by maximizing their classification power in distinguishing responders from non-responders in the pre-treatment phase. Together, these steps aim to infer either a set of "resistance downregulated interactions" (RDI) or "resistance upregulated interactions" (RUI). When multiple training cohorts are available, the training cohorts used in step 1 and step 2 are mutually exclusive and iteratively exchanged resulting in an ensemble model with multiple sets of inferred RDIs or RUIs for a given testing-cohort.

## Differential activation analysis in step 1

The first step of IRIS aims to determine interactions that are either "upregulated" or "downregulated" following the emergence of immune checkpoint blockade (ICB) resistance. This involves using cell-type-specific ligand-receptor interactions as features, and the objective is to identify differentially activated interactions between pre-treatment and non-responder (RECIST criteria: stable/progressive disease) post-treatment samples. This step both establishes the directionality of the interactions following ICB resistance and reduces the search space for the subsequent step 2.

Training involves utilizing the cell-type-specific ligand-receptor interaction activity profile, a direct output of LIRICS. When multiple training cohorts are available, interaction activity profiles are merged, including the union of inferred interactions within each cohort. The initial feature-space can consist of up to 3776 possible ligand-receptor interactions between ten cell types within the melanoma TME, sourced from the LIRICS-curated database. A fisher's exact test is employed to identify interactions that are significantly activated (FDR < 0.2 per cell type pair) in either the post-treatment non-responder or the pre-treatment groups, categorizing the selected interactions as either resistance "upregulated" or resistance "downregulated" respectively. The output of step 1 is a ranked list (by FDR values) of interactions that are categorized as either RUIs or RDIs. These interactions compose the search space for step 2.

## Hill climbing aggregative feature selection for interactions that maximize classification of patient response to ICB in step 2

The second step of IRIS aims to identify an optimal set of interactions that are functionally relevant in dictating ICB response. This involves starting from the list of interactions (RDI or RUI) identified in the previous step as features, and the objective is to select for an optimal set of interactions that maximizes the classification power in distinguishing responders (RECIST criteria: partial/complete response) and non-responders (RECIST criteria: stable/progressive disease) in the pre-treatment phase. The method of step 2 was developed by amalgamating components of the feature selection methodologies outlined in our previous work from Wang et al[15]. and Auslander et al.[9].

The algorithm consists of an iterative procedure comprising three sequential stages. First, the pre-treatment samples in the training cohort are randomly split into three folds: two for training and one for testing. Second, starting from an empty set, the procedure greedily adds the most effective discriminating interactions (ranked by FDR by cell-pair from step 1) in a step-by-step manner until no further improvement in the classification accuracy (in terms of AUC) of ICB response is achieved within the training-folds. Considering the directional nature of individual features established in stage 1, we anticipate that for RUIs from step 1, less activations correspond to a response. Conversely, for RDIs, higher activation levels correlate with a response. Third, the selected set of interactions from the training folds *together* is evaluated in terms of classification accuracy and scored on the testing-fold. A test AUC > = 0.6 results in a reward score of 1 for all selected set of interactions, while a test AUC < 0.4 incurs a penalty score of -1. AUCs in-between and all unselected features received a score of 0. This process is repeated for 500 iterations.

Following 500 iterations, we sum the scores for each individual interaction across all iterations, termed feature score. We identify the features with the greatest feature score from the entire 500 solutions, which approximates a solution that minimizes the risk of overfitting. During the feature selection, it is possible to estimate the empirical $p$-value linked to each feature having a higher feature score than the feature score obtained by random chance. This estimation is accomplished by shuffling the values of each solution derived from the hill climbing feature selection, repeating this procedure 1000 times for each iteration. By performing 1000 random permutations of 500 solutions, a null distribution representing the feature score via random chance is constructed. We selected features with an empirical $p$-value < 0.05 as either our RDIs or RUIs depending on the input features directionality. The empirical $p$-value for each feature j is estimated as follow:

$$p_j = \frac{1}{1000} \sum_{i=1}^{1000} 1_{\text{null feature score}_{ij} > \text{observed feature score}_{ij}} \quad (1)$$

In this study, the pre-treatment cohort from Auslander et al. was excluded during step 2 feature selection, owing to the limited number of responders, which hindered the reliable determination of prediction accuracy.

## Calculating RDS and RUS

Following the identification of RDI or RUI in melanoma, the calculation of each tumor samples' resistance-downregulated score (RDS) and resistance-upregulated score (RUS) for a given tumor sample is computed as followed:

$$\text{RDS or RUS} = \frac{\sum_{i=1}^{n} \frac{\text{number of active inferred RDI or RAI}_i}{\text{total number of inferred RDI or RAI}_i}}{n} \quad (2)$$

First calculate the fraction of activated inferred interactions derived from each set of RDIs or RUIs inferred (i) within the ensemble model. Then average the fraction of activated inferred interactions across all (n) sets of RDIs or RUIs within the ensemble model. The ultimate RDS and RUS scores are scaled for cross-cohort comparisons.

A merged ensemble model is also derived by combining all individual RDI ensemble models originally inferred for each cohort: Gide et al., Liu et al., Auslander et al., Riaz et al., and PUCH. This unified

merged model is applied to the two independent patient cohorts: TCGA-SKCM and Jerby-Arnon et al.

## Calculating odds ratio

To establish the proper classification threshold for distinguishing true responder and true non-responder from false responder and non-responder for each testing cohort, we considered both the set of RDIs inferred, and tumor samples extracted from each pre-treatment training cohort used in step 2 are considered. For each training cohort, the proportion of activated inferred interactions in each sample are calculated and scaled across all tumor samples. Using the scaled score in conjunction with response information for each tumor sample in the training cohort, we compute the optimal maximizing cut point using the cutpointr() function from the cutpointr package (v1.1.2) in R[56]. The final threshold is obtained by taking the mean of the individual thresholds computed using each training cohort.

## Benchmarking with other state-of-the-art ICB predictors

To benchmark IRIS' performance, we computed the accompanying score for each state-of-the-art ICB predictor in each tumor sample following the original method. The T cell exhaustion (Tex) signature was computed by averaging the expression of available genes within the signature, including *LAG3, TIGIT, PDCD1, PD1, HAVCR2, TIM3*, and *CTLA4*. The TIDE score was computed using the online tool (http://tide.dfci.harvard.edu). The resF signature was computed using the author's original published code available from (https://github.com/livnatje/ImmuneResistance). For Melanocytic plasticity score (MPS)[10], IFNG signature[57], Cytotoxic signature[11], T cell inflamed GEP (Tin)[58], and IMPRES[9], we computed the signature scores following the exact methods detailed in the original publication.

## Spatial RNA-seq dataset

We collected two publicly available metastatic melanoma spatial transcriptomics datasets from Thrane et al[37]. and Biermann et al.[36], acquired from the legacy and SlideSeqV2 platforms respectively. Thrane et al. includes slides of treatment naïve melanoma lymph node metastasis (n = 8). Biermann et al. includes slides of treatment naïve melanoma brain metastasis (n = 9), treatment naïve extracranial melanoma metastasis (n = 2), and anti-CTLA4 post-treatment extracranial melanoma metastasis (n = 2). The two post-treatment samples were derived from non-responder (RECIST criteria: progressive disease) patients to the ICB regimen. Slides MBM07.1, MBM13.1, and ECM08.1 from Biermann et al. were excluded from our analyzes due to insufficient spatial coordinate coverage or prior adoptive T cell therapy. Additionally, we collected all single-nuclei RNA-seq (snRNA-seq) data from Biermann et al. of profiled metastatic melanoma samples (n = 28). Legacy data from Thrane et al. was downloaded from (https://www.spatialresearch.org/resources-published-datasets/), while SlideSeqV2 and snRNA-seq data for Biermann et al. was downloaded from GEO (GSE185386).

## Aligning single-cell to spatial transcriptomics using CytoSPACE

To infer the activity of ligand-receptor interactions in spatial data using SPECIAL (see below), we first deconvolved spatial transcriptomics (ST) data to single-cell resolution utilizing CytoSPACE (v1.0.6)[38] (https://github.com/digitalcytometry/cytospace). CytoSPACE requires two inputs: 1. count matrices for a spatial transcriptomic slide, and 2. count matrices for a reference single-cell transcriptomics dataset with cell type annotations. It outputs deconvolved spatial transcriptomics data, assigning spatial coordinates for selected individual single-cells from the references scRNA-seq dataset. The cell abundance within each coordinate of the spatial transcriptomic slide for each cell type is computed as the fraction of cells within that specific region.

For bulk ST data from Thrane et al. (using the legacy platform), we assigned spatial spots with scRNA-seq data (in counts) of treatment naïve samples (n = 16) from Jerby-Arnon et al.[12] as the reference. The

cell types were limited to the ten specified cell types mentioned earlier in our single-cell analysis. To optimize CytoSPACE performance for smart-seq counts data from Jerby-Arnon et al., we estimated cell fraction composition for each slide using Spatial Seurat[59] as user-provided cell fraction input for CytoSPACE. Additionally, we set the geometry as square, down-sampling off, and mean cell numbers to 20 following the guidance of Vahid et al. when applying CytoSPACE to legacy spatial platforms and smart-seq reference data[38].

For SlideSeqV2 platform (i.e., single-cell resolution ST) from Biermann et al., we first assigned each spatial spots with individual cell types using the RCTD[60] pipeline in doublet mode from the spacexr package in R. Each SlideSeqV2 slide was annotated using all metastatic melanoma snRNA-seq data (n = 28) with cell type annotations. The cell types from the snRNA-seq data were manually reannotated as shown in Supplementary Table 1. Doublets, "low-quality" cells, and cycling cells were excluded. The output of RCTD included the first cell type annotation for all spatial spots and the second cell type annotation only if RCTD inferred the spatial spot as a "doublet certain". The RCTD framework was adopted from Biermann et al.'s methods[36]. Next, we executed CytoSPACE (single-cell mode) twice for each SlideSeqV2 slide from Biermann et al. using only the matched patient snRNA-seq data as a reference. The first run incorporated all spatial spots with their respective first cell type annotation from RCTD as user-provided input, while the second run was limited to "doublet certain" spatial spots with their respective second cell type annotation. Cell types absent in either the matched patient spatial transcriptomics slide or snRNA-seq were excluded before running CytoSPACE.

## Inferring activated cell-type-specific ligand-receptor interaction from single-cell transcriptomics aligned to spatial transcriptomics using SPECIAL

To quantify the activity of cell-type-specific ligand-receptor interactions within each spatial transcriptomics slide, we further developed our in-house single-cell ligand-receptor inference tool called SOCIAL, into SPECIAL (**SP**atial transcriptomics c**E**ll-**C**ell **I**nteraction **AL**gorithm; see Supplementary Fig. 12). This novel iteration is customized specifically for spatial transcriptomics with aligned single-cell transcriptomes, the direct output of CytoSPACE (see above). The three major steps of SPECIAL are outlined below (see Supplementary Fig. 12).

In the first step, spatial coordinates for each slide are divided into spatial "regions" with a diameter set to approximately 250 µm, a distance conducive to paracrine ligand-receptor interactions based on literature[39], via two possible methods. For bulk ST platforms (i.e., Legacy platform or Visium 10X), we developed a sliding window approach. This approach consolidates cells within a specified radius at each labeled spatial coordinate. The number of regions is contingent upon the number of spatial coordinates sequenced within a given slide, and each region's spatial coverage can overlap with multiple other regions. For SlideSeqV2 platform (or similar single-cell resolution ST platforms with a circular sequenced area or "puck") we developed a K-means clustering-based approach. This method clusters individual cells based on their x and y coordinates into non-overlapping circular regions based on a specified diameter. Here, the number of regions is determined by the number of circle regions with diameter $i$ that can fit within a circular sequenced area of diameter $j$. For both approaches, regions containing a single cell type are omitted from downstream analysis.

In this study, we employed the sliding window approach to Thrane et al.'s cohort (Legacy platform), consolidating cells within a 1-unit radius (maximum 300 µm diameter region). While for Biermann et al.'s cohort (SlideSeqV2 platform), we utilized the K-means clustering approach, clustering each slide into 144 regions. We chose 144 regions based on the number of 250 µm diameter circle regions that can fit within the 3000 µm diameter circle "puck" sequenced by the SlideSeqV2 platform.

In the second step, to discern the "activated" or "inactivated" status of curated LIRICS interactions within each spatial region, we employed the same proprietary R tool, SOCIAL, described earlier. Specifically, SOCIAL is independently applied to each region, encompassing the respective single-cell transcriptomes.

In the third step, interactions with an empirical $p$-value < 0.05 and both ligand and receptor mean expression greater than the median across all regions (analogous to LIRICS) are categorized as "activated" through this process.

**Reporting summary**

Further information on research design is available in the Nature Portfolio Reporting Summary linked to this article.

## Data availability

The bulk RNA-seq data, CODEFACS' deconvolved expression and cell fraction data for Gide et al.[16], Riaz et al.[18], Liu et al.[17], and TCGA-SKCM from Wang et al[15]. are available via Zenodo repository[61] (https://zenodo.org/records/5790343). TCGA-SKCM's survival timelines are available from the UCSC Xena browser[53] (https://xenabrowser.net), and pathology classifications are available from Saltz et al.[33]. The bulk RNA-seq data for Auslander et al[9]. are available from GEO under the accession number GSE115821, and for PUCH[19] from GitHub (https://github.com/xmuyulab/ims_gene_signature). Single-cell RNA-seq data from Jerby-Arnon et al[12]. are available from GEO under the accession number GSE115978, with additional cell meta information sourced from TISCH2[62] (http://tisch.comp-genomics.org). The spatial RNA-seq data from Thrane et al[37]. are available from (https://www.spatialresearch.org/resources-published-datasets/). For Biermann et al.[36], both single-nuclei and spatial RNA-seq data are available from GEO under the accession number GSE185386. The CODEFACS, LIRICS, SOCIAL, and SPECIAL data (and relevant inputs) generated in this study have been deposited in Zenodo repository[63] (https://zenodo.org/records/13172848). Source data are provided with this paper. The remaining data are available within the Article, Supplementary Information, or Source Data file. Source data are provided with this paper.

## Code availability

The tools (IRIS, SOCIAL, and SPECIAL) and codes for reproducing the results of this study are available via GitHub (https://github.com/KWangLab/NatCommun_Sahni2024). Deconvolution tools CODEFACS and LIRICS are available via Zenodo repository (https://zenodo.org/record/5790343). CytoSPACE is available via GitHub (https://github.com/digitalcytometry/cytospace).

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

## Acknowledgements

This research was supported in part by the NIH Intramural Research Program, National Cancer Institute. This work utilized the computational resources of the NIH HPC Biowulf cluster (http://hpc.nih.gov). The results here are in part based upon data generated by the TCGA Research Network: https://www.cancer.gov/tcga. We would additionally like to acknowledge members of the Cancer Data Science Laboratory for their helpful feedback on this work. Figure 1 panel B, Fig. 3 panel D, Fig. 6, Supplementary Fig. 11, and Supplementary Fig. 12, created with BioRender.com, released under a Creative Commons Attribution-NonCommercial-NonDerivs 4.0 International license.

## Author contributions

Conceptualization of the study: K.W. and E.R. Data collection and curation: S.S. with assistance from B.W., K.W., and S.P. Software development: S.S., K.W., and S.P., supervised and conceptualized by K.W. and E.R. Selection of methods for data analysis: S.S. and K.W. Literature search of related work: S.S. and K.W. Visualization: S.S. and D.W., with assistance from K.W. Writing first draft: S.S., K.W. and E.R. Writing later

drafts and editing: S.S., K.W., E.R., S.R.D., M.N., I.F., and C.D. Supervised the study: K.W. and E.R.

## Funding

## Competing interests
E.R. is a co-founder of Medaware Ltd, Metabomed Ltd, and Pangea Biomed Ltd (divested from the latter). E.R. serves as a non-paid scientific consultant to Pangea Biomed Ltd. The rest of the authors declare that they have no potential competing interests.
