## [Peer Review File · Nature Communications]

A machine learning model reveals expansive downregulation of ligand-receptor interactions that enhance lymphocyte infiltration in melanoma with developed resistance to Immune Checkpoint BlockadeEditorial notes: Parts of this Peer Review File have been redacted as indicated to remove third-party material where no permission to publish could be obtained. The name of the handling editor has been redacted to ensure confidentiality.

REVIEWER COMMENTS

Reviewer #1 (Remarks to the Author): with expertise in cancer immunotherapy

This is a nice article by Sahni et al. describing their efforts to study ligand-receptor interactions possibly mediating resistance to immune checkpoint blockade. The article is well-written and provides a clear description of the study and results; which can drive further research in that area.

Minor comments:

- I recommend revising the title for better clarity.
- I would argue that response rates to ICB should not be described as limited. From a clinical standpoint, this remains clinically meaningful.
- Try to limit the use of abbreviations. All abbreviations in the manuscript should be written out in full on first use. Abbreviations should be added after description of the full term (not before).
- The second paragraph in introduction can be summarized, or rephrased, with most of data moved to discussion.
- Please do not present results in introduction.
- Kindly present p-values according to journal's format.

Reviewer #2 (Remarks to the Author): with expertise in bioinformatics, cancer immunology

In this manuscript, Sahni et al. develop and validate a tool called IRIS which is designed to identify cell-cell interactions which mediate resistance to ICB therapy from bulk gene expression data. Using bulk RNA-Seq data from 400 samples in 5 studies that investigated ICB resistance in melanoma, the authors perform a deconvolution method published from this lab (CODEFACS) which infers relative cell abundance, cell type specific gene expression,

and cell-cell interactions (or lack thereof), in each sample. They then evaluate the predictive capacity of the cell-cell interactions which are either active (RAS) or inactive (RDS) in the resistance condition. Overall, the manuscript has an interesting workflow and provides a useful approach to utilizing bulk RNA-Seq data; however, the results do not appear to be significant enough to merit only 3 figures of data. The primary takeaway from the IRIS model essentially recapitulates the phenomena whereby cold tumors typically do not respond to ICB, which is a well described phenomena. The RDS signal was, however, prognostic for response to ICB, and more so than a number of other published gene sets, albeit only slightly so. Beyond this result, the additional mechanisms highlighted in the text have no form of validation to support the validity of the predicted interactions nor their mechanistic significance.

Major points:

In the introduction and throughout the paper, resistance is likely overapplied to all relapsed and refractory cases. However, not all non-responders are resisting the drug. Some just have no T cells present to activate. This is not resistance. The authors assume that all non-response is resistance, but sometimes non-response is passive, meaning the ligand-receptor interaction is not mediating “resistance”. This is essentially the primary result of the paper, which diminishes the utility of the predictions. To be clear, this does not mean the methodology is inadequate, but perhaps a different type of tumor would give a result not so dependent on the hot vs. cold variance.

Considering the cell-cell interaction prediction is still prone to false positives, wouldn't it be better to select based on likelihood of being real as opposed to predictive power?

Filtering/enriching based on predictive capacity just compounds the effect of false positives. It is important to be as robust and strict at this step as possible since the result is a prediction from a predicted gene expression profile, from a predicted cell abundance from bulk data (ie. lots of degrees of freedom added).

For the other predictors used on page 7 lines 171-174, the methods do not state how these

signatures were scored. Simply using the gene set does not mean that the highest accuracy possible was obtained from that predictor. Some of the signatures may use a non-linear function to predict response.

The performance presented in the results section (Figure. 2C) lacks statistical significance. The authors provide no validative support for any of the mechanisms discussed in the results and discussion. Without even validating the expression of ligands or receptors on the cells in question, let alone the predicted interactions or their presence in resistant tumors, leaves the entire work as “signatures”, but no mechanism can be gleaned. The only validation provided is a TIL estimation via pathology, which only supports that the RDS is predictive of hot vs. cold tumors, something which was already known and supported in the literature.

Minor points:

Abbreviations are defined before they are used, for example, R and NR in Figure 1 are not defined in the legend in Figure 1. Abbreviations are repeatedly defined and there is some discrepancy in abbreviations’ definitions. For example, RDS has two definitions in the text, resistance deactivated (score) and resistance-depleted score.

There is no definition of the blue cell in figure 5.

Reviewer #3 (Remarks to the Author): with expertise in cancer immunology, ligand-receptor interactions

In this manuscript, Sahni et al. developed a machine learning model to predict potential ligand-receptor interactions that could drive or mediate resistance to immune checkpoint inhibitors. They employed a previously developed computational platform to deconvolve bulk RNA-seq samples from melanoma patients undergoing ICB therapy and infer cell type specific ligand-receptor interactions and now they extended this platform to predict which of those interactions are activated or deactivated during ICB therapy. Their results show that the deactivated interactions are more predictive of response and survival and the

authors subsequently focus on those. The quality of the data appears very good, the authors use five different cohorts of melanoma samples, as well as scRNA-seq data, the analyses are well done. The predicted ligand-receptor interactions are well known to be important for driving infiltration of immune cells, showing that the method works very well, however there is not much new biology in the manuscript, the interactions that the authors emphasise are well known interactions. The potential mechanisms of resistance, why in some tumours some interactions are deactivated, leading to poor CD8 T cell infiltration and resistance and others are not, are not investigated, as the authors also mention in their limitations.

Major comments:

It would be useful for the reader to include a brief summary of the cell-cell communication method in the main text, the authors mention they use an in-house method, but in the Methods it is mentioned that they use the Kumar et al. method which is already published.

How common are the RDI/RSI between cohorts? Are all of them important for survival/prediction of response? Why are they different between cohorts, are there any specific for a specific checkpoint inhibitor? The authors mention that the superiority of their prediction method over other is that they can identify potential targets, but how would you target so many interactions - or how would you identify the very relevant ones?

The CXCR4 - CXCL12 interaction has been described in other studies to be involved in preventing CD8 T cell infiltration in tumour, however the authors here say the opposite, could they explain more?

For the scRNA-seq data, the authors don't mention how or whether they reanalysed the data in the Methods. Did they use already annotated cells and they regroup them into global cell types (they mention they reannotated it to match the bulk data, but I could not find any methods for the analysis)?

I appreciate the reason why the authors used global cell type clusters in the scRNA-seq data,

to show that the method also works there, but they can expand these analysis and utilise the heterogeneity and granularity of cell types/states that scRNA-seq data provides to look more in depth of which more detailed cell types have expression of the predicted ligand-receptor interactions to get a more defined hypothesis of a potential mechanism.

Responses to reviewers' comments

General comments: Some of the reviewer comments were not numbered; we added numbers to aid in cross-referencing. References to papers that we cite in any of the responses are collected at the end of this document. Within this document, we employed the numbering system **RiCjFk** or **RiCjTk** to provide unique identifiers to *figures* and *tables*. Here, 'i' represents the number of the reviewer, 'j' denotes the comment number, and k is the index of the figure or table in response to that comment. The corresponding numbers in the manuscript are noted. The reviewer comments are copied verbatim in black font and alternate with our responses, which are presented in **red bold font**. Any text additions made to the manuscript to address the reviewers' comments are highlighted in **turquoise** and are indented, except for very short additions.

Reviewer #1:

This is a nice article by Sahni et al. describing their efforts to study ligand-receptor interactions possibly mediating resistance to immune checkpoint blockade. The article is well-written and provides a clear description of the study and results; which can drive further research in that area.

Thanks for your kind words and much appreciated.

1. *I recommend revising the title for better clarity?*

Response:

We appreciate the reviewer's comment. We have revised the title to “Melanoma acquires resistance to Immune Checkpoint Blockade via expansive downregulation of ligand-receptor interactions enhancing lymphocyte infiltration” in lines 1 to 2.

2. *I would argue that response rates to ICB should not be described as limited. From a clinical standpoint, this remains clinically meaningful.*

Response:

Indeed, thanks and we have revised the statement to “has room for improvement” in line 39.

3. *Try to limit the use of abbreviations. All abbreviations in the manuscript should be written out in full on first use. Abbreviations should be added after description of the full term (not before).*

Response:

Thanks. we have corrected the description of abbreviations for IRIS in lines 19-20, 78, and 106; CODEFACS in line 76; LIRICS in lines 76-77; TCGA-SKCM in lines 211-212.

We have also simplified and removed superfluous descriptions of abbreviations for TME in line 269; TCGA-SKCM in lines 324 and 327; TPM in line 522.

We also added the description of abbreviations for ICB in lines 452.

Lastly, we have removed abbreviations mentioned including LRI for ligand-receptor interactions in lines 21, 128, 131, 549, and 553; ML for machine learning in line 119; NK for natural killer cells in line 99.

4. *The second paragraph in introduction can be summarized, or rephrased, with most of data moved to discussion.*

Response:

Thanks so much for the reviewer's comment. We think that this paragraph provides a pivotal motivation for our study, listing previous relevant studies and outlining their significant limitations (of course, the limitations of our own method are appropriately then addressed in the Discussion). We have chosen to retain this content here and hope it's okay, but we are open to accommodating other suggestions.

5. *Please do not present results in introduction.*

Response:

Thanks. Accordingly, we have rephrased the final summary paragraph of the introduction section to have a more general overview tone, as outlined in lines 82-91, as follows:

“IRIS aims to discern a catalog of cell-type-specific ligand-receptor interactions, pivotal in fostering resistance to ICB therapy, leveraging data from the most influential ICB therapy cohorts in melanoma (1–5). Our findings underscore that these identified interactions hold promise as potent biomarkers for predicting ICB therapy response, surpassing previously published transcriptomic biomarkers. Moreover, functional analysis of these interactions offers profound insights into the underlying mechanisms of resistance development. Subsequently, we validated the potential functions of these interactions through comprehensive analysis of multi-modal transcriptomics datasets. In essence, our study introduces a pioneering machine learning approach for systematically identifying treatment

resistance-relevant ligand-receptor interactions. Importantly, this versatile method holds potential for application across diverse scenarios.“

6. *Kindly present p-values according to journal's format.*

Response:

Thanks. We have adjusted the presentation of our p-values to align with the journal's format, as indicated in lines 172, 178-179, 203, 205, 214-215, 231, 270-271, 271, 272-273, 273, 274, 298-299, 326, 328, 332-333, 334, 334-335, 344-345.

Additionally, we have provided the exact p-values (instead of $P < 2.2e-16$) in lines 178-179 (twice), 231, 326, 328, 332-333, 334, 334-335, 1125, 1127, 1128, 1131.

Reviewer #2:

In this manuscript, Sahni et al. develop and validate a tool called IRIS which is designed to identify cell-cell interactions which mediate resistance to ICB therapy from bulk gene expression data. Using bulk RNA-Seq data from 400 samples in 5 studies that investigated ICB resistance in melanoma, the authors perform a deconvolution method published from this lab (CODEFACS) which infers relative cell abundance, cell type specific gene expression, and cell-cell interactions (or lackthereof), in each sample. They then evaluate the predictive capacity of the cell-cell interactions which are either active (RAS) or inactive (RDS) in the resistance condition. Overall, the manuscript has an interesting workflow and provides a useful approach to utilizing bulk RNA-Seq data; however, the results do not appear to be significant enough to merit only 3 figures of data. The primary takeaway from the IRIS model essentially recapitulates the phenomena whereby cold tumors typically do not respond to ICB, which is a well described phenomena. The RDS signal was, however, prognostic for response to ICB, and more so than a number of other published gene sets, albeit only slightly so. Beyond this result, the additional mechanisms highlighted in the text have no form of validation to support the validity of the predicted interactions nor their mechanistic significance.

1. *In the introduction and throughout the paper, resistance is likely overapplied to all relapsed and refractory cases. However, not all non-responders are resisting the drug. Some just have no T cells present to activate. This is not resistance. The authors assume that all non-response is resistance, but sometimes non-response is passive, meaning the ligand-receptor interaction is not mediating “resistance”. This is essentially the primary result of the paper, which diminishes the utility of the predictions. To be clear, this does not mean the methodology is inadequate, but perhaps a different type of tumor would give a result not so dependent on the hot vs. cold variance.*

Response:

We appreciate the reviewer's comments. Resistance to immune checkpoint blockade (ICB) therapy has been broadly categorized into primary resistance and acquired resistance. Primary resistance occurs when initially patients do not respond to ICB therapy, while acquired resistance refers to cases where patients initially respond to treatment but later experience relapse (6). One mechanism contributing to primary resistance, but certainly not the only one, is the lack of T cells (6). A significant proportion of patients experience primary resistance, and distinguishing those from acquired resistance samples is challenging due to the difficulties associated with long-term follow-up. In our study, we did not differentiate between these two types of resistance due to a lack of available information. However, as noted by the reviewer, our analysis suggests that the majority of cases likely involve primary resistance. To clarify, in the introduction, we have reiterated the distinction between two pathways of ICB therapy resistance: primary and acquired. This clarification is provided in lines 40-43, as follows:

“The majority of patients receiving checkpoint therapy face the challenge of developing resistance, including both primary resistance where the tumor does not respond to treatment initially and acquired resistance where initial responding tumors turn resistant over time (6–8).”

Additionally, we clarify in the results section that we classify both progressive disease and stable disease as forms of non-response or resistance, as is conventionally done, in lines 138-139. It reads as follows:

“In this study, we classify both RECIST criteria progressive and stable disease as forms of ICB resistance or non-response.”

Lastly, we clarify in the methods section that the classification of non-responder and responder is based on RECIST criteria in lines 519-521. It reads as follows:

“Across all cohorts, we binarized patients’ ICB response to either responder (RECIST criteria: complete/partial response) or non-responder (RECIST criteria: progressive/stable disease).”

2. *Considering the cell-cell interaction prediction is still prone to false positives, wouldn't it be better to select based on likelihood of being real as opposed to predictive power? Filtering/enriching based on predictive capacity just compounds the effect of false positives. It*

is important to be as robust and strict at this step as possible since the result is a prediction from a predicted gene expression profile, from a predicted cell abundance from bulk data (ie. lots of degrees of freedom added).

Response:

Thanks - We appreciate this insightful comment and fully acknowledge the importance of prioritizing authentic interactions, particularly given our objective of identifying potential therapeutic targets. We note that the ligand-receptor interactions utilized in our analysis were derived from established knowledge rather than chosen through a de novo search, thereby enhancing the robustness of our findings and their likelihood. Specifically, we utilized our meticulously curated LIRICS database, compiled from the literature and published previously by our laboratory (9). This database encompasses a comprehensive collection of reported and validated ligand-receptor interactions, along with information on relevant cell types and the annotated biological functions of those interactions.

Furthermore, we acknowledge the reviewer's concern regarding the reliance on predicted deconvolved gene expression, which may have inherent limitations. To address this, we conducted validation using a single-cell ICB cohort, allowing for more accurate quantification of cell type-specific expression. The results of this validation, which largely confirms our previous findings, can be found in lines 222-234 of the results section. Additionally, to further validate our findings, we performed analyses using two spatial transcriptomics cohorts, as detailed in our response to your No. 5 comment. Our results demonstrate quite remarkable consistency across the bulk, single-cell, and spatial transcriptomics cohorts, providing robust support for our findings.

We have reemphasized in the introduction, results, and discussion that the cell-type-specific ligand-receptor interactions are inferred from a list of 'known' interactions curated from the literature, as indicated in lines 74-77, 111-116, and 297-299. In lines 74-77, it reads as follows:

“To dissect bulk transcriptomes and prioritize literature-curated, clinically relevant cell-cell interactions, we recently developed COncident DEconvolution For All Cell Subsets (CODEFACS) and LIgand-Receptor Interactions between Cell Subsets (LIRICS), respectively.”

In lines 111-116 it reads as follows:

“1. Literature-curated cell-type-specific ligand-receptor interaction activity profiles (denoting either activation: 1 or inactivation: 0) in each tumor sample, which is inferred using LIRICS from the deconvolved expression – an interaction is considered as activated if

the (deconvolved) expression of both its ligand and receptor genes is above their median expression values across the cohort samples, and deactivated otherwise; 2. The corresponding ICB response outcome for each patient.”

In lines 297-299 it reads as follows:

“Notably, reviewing the functional enrichment of the RDIs based on LIRICS curated annotations, there is a prominent over-representation of chemotaxis interactions (one-sided Fisher’s test $P = 0.0008$, odds ratio = 3.21; Fig. 3C).”

We also replaced the term “de-novo” to “cell-type-specific” in the introduction in lines 77-80 to resolve ambiguity, as follows:

“These methods have laid the basis for the development here of Immunotherapy Resistance cell-cell Interaction Scanner (IRIS), a supervised machine learning method for identifying cell-type-specific ligand-receptor interactions relevant to ICB therapy response in the TME.”

Lastly, we reemphasized in the discussion that IRIS does not infer de-novo interactions in line 482-483, as follows:

“Secondly, our study “*only*” considered well-defined ligand-receptor interactions from the existing literature instead of inferring de-novo cell-type-specific interactions.”

3. *For the other predictors used on page 7 lines 171-174, the methods do not state how these signatures were scored. Simply using the gene set does not mean that the highest accuracy possible was obtained from that predictor. Some of the signatures may use a non-linear function to predict response.*

Response:

We appreciate the reviewer's comments. Indeed, the methods by which state-of-the-art ICB predictors calculate response scores vary significantly. To benchmark IRIS's performance in an unbiased manner, we meticulously followed the instructions provided by the respective authors of these predictors and directly utilized their codes or web services.

To provide transparency and ensure accurate performance comparisons, we have added an additional section in the methods detailing how we replicated the calculation of other

transcriptomics based ICB predictors. This information can be found in lines 699-708, as follows:

“Benchmarking with other state-of-the-art ICB predictors: To benchmark IRIS’ performance, we computed the accompanying score for each state-of-the-art ICB predictor in each tumor sample *following the original method*. The T cell exhaustion (Tex) signature was computed by averaging the expression of available genes within the signature, including *LAG3, TIGIT, PDCD1, PD1, HAVCR2, TIM3, and CTLA4*. The TIDE score was computed using the online tool (<http://tide.dfci.harvard.edu>). The functional ICB resistance (resF) signature was computed using the author’s original published code available from (<https://github.com/livnatje/ImmuneResistance>). For Melanocytic plasticity score (MPS), IFNG signature, Cytotoxic signature, T cell inflamed GEP (Tin), and IMPRES, we computed the signature scores following the exact methods detailed in the original publication.”

4. *The performance presented in the results section (Figure. 2C) lacks statistical significance.*

Response:

Thanks. We have added statistical significance (one-sided Wilcoxon test) to the result section corresponding to Figure 2C in lines 191-195.

5. The authors provide no validative support for any of the mechanisms discussed in the results and discussion. Without even validating the expression of ligands or receptors on the cells in question, let alone the predicted interactions or their presence in resistant tumors, leaves the entire work as “signatures”, but no mechanism can be gleaned. The only validation provided is a TIL estimation via pathology, which only supports that the RDS is predictive of hot vs. cold tumors, something which was already known and supported in the literature.

Response:

We appreciate the comment. First and foremost, it is essential to underscore our primary focus: the development of a new, highly predictive biomarker signature for assessing response to ICB therapy in patients. To achieve this goal, we have devised a novel machine learning method tailored to identify key ligand-receptor interactions associated with ICB resistance in melanoma. Through this approach, we have pinpointed approximately 100 ligand-receptor interactions whose collective status serves as a robust biomarker for predicting ICB response in patients. Leveraging our previously developed resource, LIRICS, which offers annotated functionality for these interactions, we conducted a comprehensive functional enrichment analysis. This analysis uncovered RDIs’ significance in chemotaxis interactions, shedding light on their relevance to immune response modulation. Notably, this

analysis has led us to discover that tumors undergo negative selection of chemotaxis-relevant interactions on a large scale to orchestrate the TME towards resistance.

Having said that, after discussions with the editor, [editor name redacted], we have now performed additional analyses to further boost and validate our results from additional novel

computational angles. These computational validations include 1) as the reviewer requested, we have confirmed the expression of ligands and receptors involved in our identified ligand-receptor interactions within the relevant cell types; 2) we have enhanced our initial validation within a melanoma single cell ICB cohort; 3) we have bolstered our validations by utilizing publicly available spatial transcriptomics data in melanoma. We will now elaborate on each of these validation categories in the subsequent sections.

1) Validation of ligand and receptor expression in relevant cell types for RDIs:

We have confirmed the expression of ligands and receptors involved in our identified ligand-receptor interactions within the relevant cell types. To accomplish this, we extended upon our previous single-cell RNA sequencing (scRNA-seq) data from Jerby-Arnon et al., encompassing both treatment-naïve and post-ICB treated tumor samples (1). For each ligand and receptor gene within our RDI network, we conducted a one-sample t-test to assess their expression levels within the corresponding cell type. Our analysis revealed that across all cell types, nine cell-types exhibited significant (one-sample t-test FDR < 0.2) expression of over sixty percent of the inferred ligand and receptor genes (see Figure R2C5F1). Only cancer-associated fibroblasts (CAF) exhibited a lower percentage, with only 48.6% of genes showing significant expression. This observation is likely attributed to both the limited number of CAF cells sequenced and the inherent sparsity of the scRNA-seq platform.

Figure R2C5F1: Dot plots depicting the expression landscape of ligand and receptor genes within each respective cell type. The Y-axis denotes the negative log of the false discovery rate (FDR) obtained from the one-sample t-test for each ligand and receptor gene expression within each cell type. The X-axis lists the significantly expressed ligand or receptor gene within the respective cell type in ascending order. Ligand and receptor genes with an FDR < 0.2 (indicated by the dotted line) were deemed significantly expressed and are highlighted in magenta. The percentage of significant ligand and receptor genes relative to all inferred genes within the RDI network is provided in the top right corner of each plot. The respective cell type is highlighted at the top of each plot.

To integrate the new findings, we have introduced the following sentences into lines 218-222 of the Results section in the manuscript, and Figure R2C5F1 has been included as Supplementary Figure 5:

“We confirmed the expression of the ligands and receptors from our inferred RDI within the relevant cell types using single-cell transcriptomics data from Jerby-Arnon et al.’s melanoma ICB study (10). Employing a one-sample t-test for each gene within our RDI list, we evaluated expression levels within relevant cell types. Notably, across ten cell types, nine showed significant expression of over sixty percent of the inferred ligand and receptor genes (Supp Fig. 5).”

2) Enhancement of our validation within a melanoma single-cell ICB cohort:

In our initial submission, we validated our inferred RDIs from bulk cohorts using single-cell ICB cohorts from Jerby-Arnon et al.'s study. The key finding is that RDIs down-regulated in bulk resistant samples are also significantly down-regulated in single-cell resistant samples compared to the respective treatment naïve samples. In this revision, we've improved our in-house single-cell tool, SOCIAL (refer to our response to reviewer comment 1), and updated the results as outlined in lines 222-234:

“To study the role of RDIs in resistance development via single-cell transcriptomics, we further analyzed Jerby-Arnon et al.'s study by implementing both treatment response and timepoint labels. Utilizing our in-house R tool, SOCIAL (Single-cell transcriptOmics Cell-cell Interaction ALgorithm; see Supp Fig. 11), which we've developed by integrating insights from Kumar et al. (11), Vento-Tormo et al. (12), our own LIRICS framework (9), we inferred activated ligand-receptor interactions from the single-cell transcriptomics data (see Methods). Without any additional training, we directly derived the RDS score for each tumor sample using RDIs inferred from bulk ICB cohorts (Methods). The RDS scores in treatment naïve samples are significantly (one-sided Wilcoxon test $P = 1.81 \times 10^{-37}$) higher than those in post-treatment non-responding samples (Fig. 2I). Moreover, the derived RDS scores classify naïve and post-ICB resistant tumors with an AUC of 0.87 (Fig. 2J). These results further show that RDIs are predictive of ICB resistance at a single-cell resolution.”

3) validation using publicly available spatial transcriptomics in melanoma:

As previously discussed, we have identified an ensemble of approximately 100 ligand-receptor interactions that collectively serve as a robust biomarker for predicting response to Immune Checkpoint Blockade (ICB) therapy in patients. Importantly, many of these interactions exhibit nearly equal predictive power, with each individual interaction demonstrating relatively weak predictive capability on its own. Consequently, conducting specific experiments targeting individual interactions is unlikely to yield significant effects or "success" in isolation. The primary contribution of our study lies in the identification of this ensemble of interactions, which are predominantly downregulated as resistance emerges. Nonetheless, we acknowledge the necessity of providing additional independent validation for this pivotal finding and the ensemble of interactions as a whole. After consultation with the editor, [editor name redacted], we converged on performing further analysis in publicly available spatial transcriptomics, to perform a pertaining computational testing and validation on a large scale. We have conducted a thorough search and identified two

relevant spatial transcriptomics cohorts, encompassing a total of 21 melanoma slide samples, for our validation analysis. Our ensuing analysis focuses on examining the activity of our top

predicted interactions independently and their potential correlations with the extent of lymphocyte infiltration and other indicative markers of the immune state within TME. The goal of this analysis is to elucidate the correlation between our identified interactions and CD8+ T cell infiltration levels across distinct regions within each TME.

Towards this goal, we analyzed spatial transcriptomics data from metastatic melanoma samples sourced from two studies: Thrane et al. and Biermann et al. Thrane et al. comprises treatment-naïve samples (N=8) sequenced using a legacy platform, whereas Biermann et al. includes both treatment-naïve (N=11) and post-ICB resistant (N=2) samples sequenced with the SlideSeqV2 platform.

To quantify the activity of ligand-receptor interactions in each spatial transcriptomic dataset, we extended our previously developed tools (in-house single cell ligand-receptor inference tool, SOCIAL) to SPECIAL (Spatial transcriptomics Cell-cell Interaction Algorithm), a new version tailored specifically for spatial transcriptomics, leveraging the CytoSPACE tool (13). The key steps of SPECIAL are outlined as follows (please refer to the methods section for detailed procedures; see Figure R2C5F2):

- 1. Alignment between reference single-cell and spatial transcriptomic data using CytoSPACE (13) to deconvolve mixtures within each region to single-cell resolution.**
- 2. Employing a sliding window or k-means clustering approach to iteratively select a region within approximately 250 μm in diameter, a distance suggested as the range over which cells may interact with each other (14).**
- 3. Within each selected spatial region, we utilized our in-house cell-cell interaction inference tool, SOCIAL, using single-cell data.**
- 4. Comparison of the inferred cell-cell interactions within each selected region across all regions. Only the ligand-receptor pairs exhibiting higher expression (exceeding the median) for both ligand and receptor are denoted as significantly activated ligand-receptor interactions.**

[figure redacted]

Figure R2C5F2: Overview of SPECIAL. The SPECIAL input is the aligned single-cell transcriptomics to spatial transcriptomics (inferred by applying CytoSPACE to single-cell and spatial transcriptomics). It consists of three major steps: Step I utilizes either a sliding region or k-means clustering approach on bulk (i.e. Visium 10X and Legacy) and SlideSeqV2 spatial transcriptomics, respectively, to divide spatial slides into “regions” of approximately 250 μm in diameter. Step II employs SOCIAL steps 1 through 3 to infer cell-type-specific interaction activity within each ~ 250 μm region. Step III, ligand-receptor interactions are further denoted as significantly activated if the average expression levels of both the ligand and receptor genes within the respective cell type is greater than the median across all regions. The final output of SPECIAL is a cell-type-specific ligand-receptor interaction activity profile across all regions in a spatial transcriptomics slide.

It's noteworthy that following step 1, we obtained quantified cell abundance within each region on the slide. The application of our SPECIAL thus enabled us to explore the association between ligand-receptor interaction activity and CD8⁺ T cell fractions across regions. We plotted the levels of both ligand-receptor activity and CD8⁺ T cell fractions in each spatial location for each slide. We quantified the concordance between the two measurements (the level of ligand-receptor activity and CD8⁺ T cell fractions) for matched slides by utilizing the AUC metric after binarizing CD8⁺ T cell fraction across regions, considering the sparsity of both quantities. Remarkably, we observed a high level of concordance between the two (AUCs: median=0.77; sd=0.18 for Thrane et al. and median=0.62; sd=0.14 for Biermann et al.; see Fig. R2C5F3 A-B and Fig. R2C5F5 A-B) across the two cohorts. Biopsy 3 (labeled as mel3_rep1 in Fig. R2C5F3 A and B) underperforms due to the absence of any lymphoid tissue, as indicated by both manual and computational annotations in the original study (15). Additionally, we quantitatively demonstrate that spatial regions exhibiting CD8⁺ T cell infiltration displayed significantly more activated Resistance Downregulated Interactions (RDIs) compared to regions lacking infiltration across both slides (Fig. R2C5F3 A-B; Fig. R2C5F5 A-B; Fig. R2C5F6 A-B) and patient-specific resolutions (Fig. R2C5F3 C and Fig. R2C5F3 E) in both spatial cohorts. Furthermore, spatial regions from biopsied slides showcasing CD8⁺ T cell infiltration (N=4) harbored significantly

more activated RDIs compared to slides with CD8+ T cell deserts (N=4) from the Thrane et al. study (one-sided Wilcoxon test, $P = 3.27 \times 10e-8$; Figure 2D). These findings collectively underscore the role of RDIs in enhancing CD8+ T cell infiltration when activated at a spatial resolution.

We have now included Figure R2C5F3 as Figure 5 in lines 406-420 of the Results section, Figure R2C5F4 as Supplementary Figure 7 in lines 1135-1138, Figure R2C5F5 as Supplementary Figure 8 in lines 1140-1145, and Figure R2C5F6 as Supplementary Figure 9 in lines 1149-1152. Additionally, Table R2C5T1 has been incorporated as Supplementary Table 2 in lines 1184-1185. And we have included the description of these results in the manuscript, located in lines 361-390. It reads as follows:

“RDIs are associated with CD8+ T cell fraction across spatial regions within the TME

To further elucidate the role of RDIs in modulating CD8+ T cell fraction across distinct regions of individual TMEs, we analyzed spatial transcriptomics data of metastatic melanoma samples sourced from both Thrane et al. and Biermann et al. studies (15,16). Thrane et al. comprises of treatment naïve samples sequenced from legacy platform, while Biermann et al. includes both treatment naïve and post-ICB resistant samples sequenced using the SlideSeqV2 platform. To quantify region specific activity of our inferred ligand-receptor interactions within each spatial transcriptomic slide, we extended our previously mentioned in-house R tool, SOCIAL to SPECIAL (SPatial transcriptomics cELL-Cell Interaction ALgorithm; Supp Fig. 12) for spatial transcriptomics by leveraging CytoSPACE (13). In brief, we first aligned a reference single-cell and spatial transcriptomics data to deconvolve the mixtures within each region, achieving single-cell resolution using CytoSPACE. Utilizing a sliding window (or clustering) approach, we segmented the tumor slides into ~250 μm diameter regions, a distance conducive to a paracrine ligand-receptor interaction between individual cells (14). Lastly, SOCIAL inferred the activity of individual cell-type-specific interactions within each spatial region (see Methods). We quantified the concordance between the estimated levels of RDI activity and CD8+ T cell fraction for each location on every slide. This involved utilizing the AUC metric after binarizing the CD8+ T cell fraction across regions, considering the sparsity of both quantities. Remarkably, we observe high levels of concordance (AUCs: median = 0.77; sd = 0.18 for Thrane et al. and median = 0.62; sd = 0.14 for Biermann et al.) between both measurements across cohorts and platforms (Fig. 5A-B; Supp Fig. 8A-B). Biopsy 3 (labeled as mel3_rep1 in Fig. 5A-B) likely underperforms due to the absence of any lymphoid tissue, as indicated by both manual and computational annotations in the original study (15). Additionally, we quantitatively demonstrated that spatial regions with CD8+ T cell infiltration exhibited significantly (one-sided Wilcoxon test) more activated RDIs compared to regions lacking infiltration across both slide-specific (Supp Fig. 9A-B) and patient-specific resolutions (Fig. 5C and 5E) in both

spatial cohorts. Furthermore, spatial regions from biopsied slides with CD8+ T cell infiltration (N=4) harbored significantly (one-sided Wilcoxon test, $P = 3.27 \times 10^{-8}$; see Fig. 5D) more activated RDIs compared to spatial slides with CD8+ T cell deserts (N=4) from Thrane et al.'s study. These findings collectively underscore the role of RDIs in enhancing CD8+ T cell infiltration when activated at a spatial resolution.”

Now we have incorporated Figure R2C5F2 as Supplementary Figure 12 in lines 1171-1180.

We have included the description of these methods in the manuscript, located in lines 710-795. It reads as follows:

“Spatial RNA-seq dataset: We collected two publicly available metastatic melanoma spatial transcriptomics datasets from Thrane et al. (15) and Biermann et al. (16), acquired from both legacy and SlideSeqV2 platforms respectively. Thrane et al. includes slides of treatment naïve melanoma lymph node metastasis (n=8). Biermann et al. includes slides of treatment naïve melanoma brain metastasis (n=9), treatment naïve extracranial melanoma metastasis (n=2), and anti-CTLA4 post-treatment extracranial melanoma metastasis (n=2). The two post-treatment samples were derived from non-responder (RECIST criteria: progressive disease) patients to the ICB regimen. Slides MBM07.1, MBM13.1, and ECM08.1 from Biermann et al. were excluded from our analyses due to insufficient spatial coordinate coverage or prior adoptive T cell therapy. Additionally, we collected all single-nuclei RNA-seq (snRNA-seq) data from Biermann et al. of profiled metastatic melanoma samples (n=28). Legacy data from Thrane et al. was downloaded from (<https://www.spatialresearch.org/resources-published-datasets/>), while SlideSeqV2 and snRNA-seq data for Biermann et al. was downloaded from GEO ([GSE185386](https://www.ncbi.nlm.nih.gov/geo/query/acc.cgi?acc=GSE185386)).

Aligning single-cell to spatial transcriptomics using CytoSPACE: To infer activity of ligand-receptor interactions in spatial data using SPECIAL (see below), we first deconvolved spatial transcriptomics (ST) data to single-cell resolution utilizing CytoSPACE (13) (<https://github.com/digitalcytometry/cytospace>). CytoSPACE requires two inputs: 1. count matrices for a spatial transcriptomic slide, and 2. count matrices for a reference single-cell transcriptomics dataset with cell type annotations. It outputs deconvolved spatial transcriptomics data, assigning spatial coordinates for selected individual single-cells from the reference scRNA-seq dataset. The cell abundance within each coordinate of the spatial transcriptomic slide for each cell type is computed as the fraction of cells within that specific region.

For bulk ST data from Thrane et al. (using the legacy platform), we assigned spatial spots with scRNA-seq data (in counts) of treatment naïve samples (n=16) from Jerby-Arnon et al. (10) as the reference. The cell types were limited to the ten specified cell types mentioned

earlier from our single cell analysis. To optimize CytoSPACE performance for smart-seq counts data from Jerby-Arnon et al., we estimated cell fraction composition for each slide using Spatial Seurat (17) as user-provided cell fraction input for CytoSPACE. Additionally, we set the geometry as square, down-sampling off, and mean cell numbers to 20 following the guidance of Vahid et al. when applying CytoSPACE to legacy spatial platforms and smart-seq reference data (13).

For SlideSeqV2 platform (i.e. single-cell resolution ST) from Biermann et al., we first assigned each spatial spots with individual cell types using the RCTD (18) pipeline in doublet mode from the spacexr package in R (<https://github.com/dmcable/spacexr>). Each SlideSeqV2 slide was annotated using all metastatic melanoma snRNA-seq data (n=28) with cell type annotations. The cell types from the snRNA-seq data were manually reannotated as shown in Supplementary Table 2. Doublets, “low-quality” cells, and cycling cells were excluded. The output of RCTD included the first cell type annotation for all spatial spots and the second cell type annotation only if RCTD inferred the spatial spot as a “doublet certain”. The RCTD framework was adopted from Biermann et al.’s methods (16). Next, we executed CytoSPACE (single-cell mode) twice for each SlideSeqV2 slide from Biermann et al. using only the matched patient snRNA-seq data as reference. The first run incorporated all spatial spots with their respective first cell type annotation from RCTD as user-provided input, while the second run was limited to “doublet certain” spatial spot with their respective second cell type annotation. Cell types absent in both the matched patient spatial transcriptomics slide and snRNA-seq were excluded before running CytoSPACE.

Inferring activated cell-type-specific ligand-receptor interaction from single-cell transcriptomics aligned to spatial transcriptomics using SPECIAL: To quantify the activity of cell-type-specific ligand-receptor interactions within each spatial transcriptomics slide, we further developed our in-house single-cell ligand-receptor inference tool called SOCIAL, into SPECIAL (SPatial transcriptomics cell-Cell Interaction ALgorithm; see Supp Fig. 12). This novel iteration is customized specifically for spatial transcriptomics with aligned single-cell transcriptomes, the direct output of CytoSPACE (see above). The three major steps of SPECIAL are outlined below (see Supp Fig. 12).

In the first step, spatial coordinates for each slide are divided into spatial “regions” with a diameter set to approximately 250 μm , a distance conducive to paracrine ligand-receptor interactions based on literature (14), via two possible methods. For bulk ST platforms (i.e. Legacy platform or Visium 10X), we developed a sliding window approach. This approach consolidates cells within a specified radius at each labeled spatial coordinates. The number of regions is contingent upon the number of spatial coordinates sequenced within a given slide, and each region’s spatial coverage can overlap with multiple other regions. For SlideSeqV2 platform (or similar single-cell resolution ST platforms with a circular

sequenced area or “puck”) we developed a K-means clustering-based approach. This method clusters individual cells based on their x and y coordinates into non-overlapping circular regions based on a specified diameter. Here, the number of regions is determined by the number of circle regions with diameter i that can fit within a circular sequenced area of diameter j . For both approaches, regions containing a single cell type are omitted from downstream analysis.

In this study, we employed the sliding window approach to Thrane et al.’s cohort (Legacy platform), consolidating cells within a 1-unit radius (maximum 300 μm diameter region). While for Biermann et al.’s cohort (SlideSeqV2 platform), we utilized the K-means clustering approach, clustering each slide into 144 regions. We chose 144 regions based on the number of 250 μm diameter circle regions that can fit within the 3000 μm diameter circle “puck” sequenced by the SlideSeqV2 platform.

In the second step, to discern the “activated” or “inactivated” status of curated LIRICS interactions within each spatial region, we employed the same proprietary R tool, SOCIAL, described earlier. Specifically, SOCIAL is independently applied to each region, encompassing the respective single-cell transcriptomes.

In the third step, interactions with an empirical p-value < 0.05 and both ligand and receptor mean expression greater than the median across all regions (analogous to LIRICS) are categorized as “activated” through this process.”

Finally, we set to explore the correlation between the activity of RDIs and the outcomes of ICB therapy, utilizing these spatial transcriptomic ICB cohorts. Notably, Biermann et al.’s spatial cohort comprises both treatment-naïve (11 samples) and post-ICB resistant samples (2 samples), whereas Thrane et al.’s cohort includes only treatment-naïve samples. To still do our best to address this question given these data limitations, we pursued two strategies: firstly, we compared the level of ligand-receptor activity between treatment-naïve samples and post-ICB resistant samples across regions and patients using Biermann et al.’s cohort. Our analysis revealed that regions from resistant tumors exhibited significantly lower levels of activated RDIs compared to those from treatment-naïve tumors (one-sided Wilcoxon test, $P = 0.0216$; see Fig. R2C5F3 F). This trend persisted at a per-patient level, where resistant patients ($N=2$) harbored less activated RDIs across all regions compared to treatment-naïve patients ($N=6$) (one-sided Wilcoxon test, $P = 0.14$; see Fig R2C5F3 G). Additionally, we introduced another approach: for treatment-naïve samples from Thrane et al., we treated each region as a surrogate patient mixture and evaluate the surrogate patients using six state-of-the-art ICB predictors. The aim was to assess the association between RDI activity and these ICB predictors, which suggests the potential effectiveness of RDIs in predicting ICB therapy response. Our findings revealed that spatial regions with high levels of RDI activity

demonstrated significantly higher ICB signature scores, indicative of ICB response, compared to regions lacking RDI activity at a per-patient level (see Fig R2C5F7). Collectively, these findings underscore both the deactivation of RDIs during the development of ICB resistance and the functional significance of their activity in predicting ICB response at a spatial resolution.

Now we have incorporated Figure R2C5F7 as Supplementary Figure 10 in lines 1154-1159.

We have incorporated the description of these new findings into the manuscript, appearing in lines 392-405. It reads as follows:

“Furthermore, to investigate the role of RDIs in the development of ICB resistance using spatial transcriptomics, we initially compared treatment-naïve and post-ICB resistant samples from Biermann et al.’s cohort. Our analysis revealed that spatial regions from resistant tumors had significantly (one-sided Wilcoxon test, $P = 0.0216$; see Fig. 5F) lower levels of activated RDIs compared to regions within treatment naïve tumor slides. The trend persisted at a per-patient level (one-sided Wilcoxon test, $P = 0.14$; see Fig. 5G), where resistant patients (N=2) harbored fewer activated RDIs across all regions compared to treatment-naïve patients (N=6). Additionally, we analyzed treatment naïve samples from Thrane et al. by scoring each region using six state-of-the-art ICB predictors directly from the spatial transcriptomics expression. Remarkably, we observed that spatial regions with RDI activity presented significantly higher ICB signature score, indicative of ICB response, compared to regions without RDI activity at a per patient level (Supp Fig. 10). These findings collectively underscore both the downregulation of RDIs during the development of ICB resistance and the functional importance of their activity in predicting ICB response at a spatial resolution.”

[figure redacted]

Figure R2C5F3: Association between RDIs activity and T cell infiltration measures within the TME at spatial resolution. (A) Scatter plot illustrating the number of activated RDIs (scaled) within each spatial region of four spatial transcriptomics slides from Thrane et al. The X-axis corresponds to the x-coordinate of the spatial region, and the Y-axis corresponds to the y-coordinate of the spatial region. (B) Corresponding scatter plots depicting the fraction of CD8⁺ T cell infiltration (scaled) within each spatial region of the four

slides from Thrane et al. **(C)** Boxplot demonstrating the total number of activated RDIs between spatial regions with and without CD8+ T cell infiltration at a patient level in Thrane et al. **(D)** Violin plot illustrating the total number of activated RDIs within spatial regions from slides with and without any CD8+ T cell infiltration in Thrane et al. **(E)** Boxplot showcasing the total number of activated RDIs between spatial regions with and without CD8+ T cell infiltration at a patient level in Biermann et al. **(F)** Violin plot displaying the total number of activated RDIs within spatial regions from treatment-naïve and resistant/progressive disease (PD) patients to ICB in Biermann et al. **(G)** Violin plot indicating the total number of activated RDIs within treatment-naïve and resistant/progressive disease (PD) patients to ICB in Biermann et al.

[figure redacted]

Figure R2C5F4: Scatter plot illustrating the number of activated RDIs (scaled) within each spatial region of eight spatial transcriptomics slides from Thrane et al. The X-axis corresponds to the x-coordinate of the spatial region, and the Y-axis corresponds to the y-coordinate of the spatial region.

[figure redacted]

Figure R2C5F5: (A) Scatter plot illustrating the number of activated RDIs (scaled) within each spatial region of thirteen spatial transcriptomics slides from Biermann et al. The X-axis corresponds to the x-coordinate of the spatial region, and the Y-axis corresponds to the y-coordinate of the spatial region. (B) Corresponding scatter plots depicting the fraction of CD8⁺ T cell infiltration (scaled) within each spatial region of the thirteen slides from Biermann et al.

[figure redacted]

Figure R2C5F6: (A) Boxplot depicting the total number of activated RDIs between spatial regions with and without CD8+ T cell infiltration at a slide level in Thrane et al. (B) Boxplot depicting the total number of activated RDIs between spatial regions with and without CD8+ T cell infiltration at a slide level in Biermann et al.

[figure redacted]

Figure R2C5F7: Box plot illustrating the distribution of six ICB signature scores between spatial regions with and without any RDIs present at a patient level from Thrane et al. The signatures evaluated in each panel are: Cytotoxic signature (19), IFNG signature (20), IMPRES (1), MPS (21), functional ICB resistance (resF) (10), and T cell inflamed GEP (Tin) (22). The original MPS score was adjusted by multiplying the score by negative one so that greater adjusted MPS scores are associated with ICB responders for the purpose of visualization.

Original Cell Type	Reannotated Cell Type
Plasma cells	Bcell
Tumor cells	Mal
Dendritic cells	skinDC
Microglia	Macrophage
MDM	Macrophage
Monocytes	Additional-Immune
Tregs	Additional-Immune
CD8+ T cells	TCD8
B cells	Bcell
NK cells	NK
CD4+ T cells	TCD4
CNS cells	CNS
Stromal cells	CAF
Endothelial cells	Endo
Mast cells	Additional-Immune
Epithelial cells	Epithelial

Table R2C5T1: Manual reannotation of cell types from Biermann et al.'s snRNA-seq cohort.

6. *Abbreviations are defined before they are used, for example, R and NR in Figure 1 are not defined in the legend in Figure 1. Abbreviations are repeatedly defined and there is some discrepancy in abbreviations' definitions. For example, RDS has two definitions in the text, resistance deactivated (score) and resistance-depleted score.*

Response:

Thanks. Following reviewer 1's comment 3, we have addressed redundant abbreviations and descriptions. Specifically, we have refined the meaning of RDS/RDI to exclusively denote resistance downregulated score/interactions in lines 581 and 674. Additionally, we have included a description of the abbreviations R and NR in Figure 1, as outlined in line 144. It reads as follows:

“The IRIS input includes cell-type-specific ligand-receptor interaction activity profiles (inferred by applying CODEFACS and LIRICS on the tumor transcriptomics) and treatment response labels: responder (R) and non-responder (NR).”

7. *There is no definition of the blue cell in figure 5.*

Response:

Thanks. Now we have added annotations for all the “Cells” in the Figure 5. Additionally, to further illustrate and clarify, we revised the original figure 5 as Figure R2C7F1, which has been incorporated as Figure 6 in our revised manuscript. We are open to relocating this figure as a supplementary figure if it aligns better with the reviewer's preferences.

[figure redacted]

Figure R2C7F1. Hypothesized evolution of immune microenvironment in melanoma during ICB treatment. Based on our results, we propose a model of melanoma resistance following the initial response to ICB. In the pre-treatment stage, T cells are recruited to the tumor microenvironment by activated RDIs (purple curve, ligand-receptor pairs identified in this study) between various cell types (e.g. macrophages, DCs, etc.). However, the immune checkpoint interactions (e.g. PD1-PDL1) are preserved, reducing the ratio of effector T cells (green curve), allowing tumor cells to expand (yellow curve). In the post-treatment stage, the ratio of effector T cells increases due to blocked checkpoint interactions, resulting in the initial response to reduce tumor volume. In the resistance stage, as a mechanism of overcoming checkpoint blockade, RDI activities are declined resulting in decreased T cell recruitment. The progressive decline leads to diminish effector T cell population and therefore allowing tumor recurrence.

Reviewer #3:

In this manuscript, Sahni et al. developed a machine learning model to predict potential ligand-receptor interactions that could drive or mediate resistance to immune checkpoint inhibitors. They employed a previously developed computational platform to deconvolve bulk RNA-seq samples from melanoma patients undergoing ICB therapy and infer cell type specific ligand-receptor interactions and now they extended this platform to predict which of those interactions are activated or deactivated during ICB therapy. Their results show that the deactivated interactions are more predictive of response and survival and the authors subsequently focus on those. The quality of the data appears very good, the authors use five different cohorts of melanoma samples, as well as scRNA-seq data, the analyses are well done. The predicted ligand-receptor interactions are well known to be important for driving infiltration of immune cells, showing that the method works very well, however there is not much new biology in the manuscript, the interactions that the authors emphasise are well known interactions. The potential mechanisms of resistance, why in some tumours some interactions are deactivated, leading to poor CD8 T cell infiltration and resistance and others are not, are not investigated, as the authors also mention in their limitations.

Thanks for the thorough and overall positive and encouraging evaluation of our work.

- 1. It would be useful for the reader to include a brief summary of the cell-cell communication method in the main text, the authors mention they use an in-house method, but in the Methods it is mentioned that they use the Kumar et al. method which is already published.*

Response:

Thanks. Concerning the in-house method, we developed an R code called SOCIAL (Single-cell Transcriptomics Cell-cell Interaction Algorithm) to extract significant ligand-receptor interactions between any two specific cell types of interest, drawing inspiration from Kumar et al.'s (11), Vento-Tormo et al.'s (12) methodologies and our own LIRICS framework (9). We opted to create SOCIAL for four reasons: (1) to leverage the strengths of previously published methods, (2) to implement a unified R-based solution (given that the first method lacked publicly accessible code and the second was in Python), (3) to incorporate our comprehensive ligand-receptor interaction database (LIRICS), which offers uniquely annotated functions for these interactions, and (4) to accommodate for variations in ligand-receptor activity observed across patients.

SOCIAL (see Fig. R3C1F1) accepts as input single-cell transcriptomics data with cell type annotation and provides inferred significant cell-type-specific ligand-receptor interactions as its output. It specifically comprises three key steps: 1. Initially querying the LIRICS database to identify plausible ligand-receptor interactions. 2. Computing the ligand-receptor interaction score by multiplying the average expression levels of the ligand and receptor

complexes for each interaction pair and cell type. 3. Performing permutation tests (100 iterations in our study) by shuffling cell type labels to derive empirical p -values. A low p -value indicates a higher likelihood of the tested ligand-receptor interaction occurring between the two specific cell types. 4. Optionally, ligand-receptor interactions can be denoted as significantly activated if the average expression level of both the ligand and receptor genes is greater than the median across all samples.

[figure redacted]

Figure R3C1F1: Overview of SOCIAL. The SOCIAL input includes both the single-cell transcriptomics expression data and annotated cell type information. It consists of three major steps: Step I query the LIRICS database to identify plausible ligand-receptor interactions. Step II computes an interaction score by multiplying the average expression levels of the ligand and receptor complexes for each interaction pair and cell type. Step III performs a permutation test by shuffling cell type labels to derive an empirical p -value. Optionally in Step IV, ligand-receptor interactions can be further denoted as significantly activated if the average expression level of both the ligand and receptor genes within the respective cell type is greater than the median across all samples. The final output of SOCIAL is a cell-type-specific ligand-receptor interaction activity profile across all samples.

To provide further clarification, we have added Figure R3C1F1 as supplementary figure 11 in lines 1161-1169, and included the following sentence in lines 225-228 of the results section of the manuscript:

“Utilizing our in-house R tool, SOCIAL (Single-cell transcriptomics Cell-cell Interaction Algorithm; see Supp. Figure 11), which we've developed by integrating insights from Kumar et al. (11), Vento-Tormo et al. (12), and our own LIRICS framework (9), we inferred activated ligand-receptor interactions from the single-cell transcriptomics data (see Methods).”

We have incorporated a section detailing this method in lines 555-579 of the Methods section, outlined as follows:

“Inferring activated cell-type-specific ligand-receptor interaction from single-cell transcriptomics using SOCIAL: We developed an R code, SOCIAL (Single-cell transcriptOmics Cell-cell Interaction ALgorithm), to identify significant ligand-receptor interactions between two specific cell types, drawing upon insights from Kumar et al.’s (11), Vento-Tormo et al.’s (12), and our own LIRICS framework (9). Our decision to create our own code stemmed from four primary motivations: 1. Leveraging the strengths of previous methods: By combining aspects of the three approaches, we aimed to maximize the accuracy and robustness of our ligand-receptor interaction predictions. 2. Implementing an R-based solution: While the first method lacked publicly accessible code and the second was in Python, we sought to create an R-based solution for accessibility and ease of use. 3. Incorporating our comprehensive database: Our ligand-receptor interaction database (LIRICS) provided rich and informative annotations, enhancing the depth of our analysis. 4. Accommodating variations in ligand-receptor interaction activity observed across patients.

SOCIAL (see Supp Fig. 11) comprises three main steps: 1. Querying the LIRICS database: Initially, we queried the LIRICS database to identify plausible ligand-receptor interactions; 2. Computing interaction scores: Next, we computed the ligand-receptor interaction score by multiplying the average expression levels of the ligand and receptor complexes for each interaction pair and cell type. 3. Permutation testing: Following that, we performed permutation tests (utilizing 100 iterations in our study) by randomly shuffling cell type labels. This allowed us to derive empirical p -values by calculating the fraction of permutation tests resulting in a higher interaction score than the foreground score determined in step 2. A lower p -value suggests a higher likelihood of the interaction occurring. 4. Optionally, ligand-receptor interactions can be further denoted as significantly activated if the average expression level of both the ligand and receptor genes is greater than the median across all samples.”

Furthermore, we have revised the sentences in the subsequent section, found in lines 596-601, as follows:

“We employed our proprietary R tool, SOCIAL, as discussed earlier, to identify activated ligand-receptor interactions. Interactions meeting the criteria of an empirical p -value < 0.05 and both ligand and receptor mean expression levels exceeding the median across all samples (similar to LIRICS methodology) were classified as "activated" through this procedure.”

2. *How common are the RDI/RSI between cohorts?*

Response:

We thank the reviewer for this thought-provoking comment. To quantify the prevalence of RDIs across cohorts (totally 5), we specifically counted the number of RDIs significantly identified in at least M cohort models (M = 2, 3, 4, 5). For each test, we evaluated whether the number of common RDIs was significantly higher than expected using Fisher's test. The background is the number of common interactions between randomly selected interaction lists. We found that RDIs shared in two or more models (N=122, $P = 3.49 \times 10^{-38}$), three or more models (N=51, $P = 2.97 \times 10^{-21}$), and four or more models (N=12, $P = 1.091 \times 10^{-5}$) were statistically significant, suggesting a strong overlap of our inferred RDIs across cohort models. However, RDIs shared across all five models were not statistically significant (N=3, $P = 0.059$). These results testify that our identified RDIs are robust, despite the high level of tumor heterogeneity.

We have introduced a new column in Supplementary Table 1 to outline the number of cohort-specific models in which the RDI was inferred, denoted as “# of cohort models”. Additionally, our analysis has rectified corrected the count of RDIs shared across two or more cohort models to 122. The previous estimate of 134 did not account for duplicate RDIs inferred. We have revised the main text in lines 264 and 310 and rectified Figure 3A accordingly. Moreover, we have added the result for the test on common interactions in at least two cohorts to the Results section in lines 263-267. It reads as follow:

“Across cohort specific models, we find significant (one-sided Fisher’s test $P = 3.49 \times 10^{-38}$) overlap of 122 RDIs (out of 299 total inferred RDIs, see Supp. Table 1 for complete RDI network) in at-least two or more cohort models that we have studied (Fig. 3A), suggesting that IRIS inferences are robust despite the high levels of heterogeneity across patient samples.”

3. *Are all of them important for survival/prediction of response?*

Response:

Thanks – indeed, not all inferred individual RDIs are individually important for ICB (Immune Checkpoint Blockade) response and survival, as we evaluated the collective impact of these RDIs. Our analyses for ICB response (outlined in lines 174-179 and depicted in Supp. Figure 2A-B) revealed that 173 (58%) and 139 (46%) of RDIs were significantly activated in responder tumors compared to non-responder tumors in both pre-treatment and post-treatment samples, respectively. Similar analyses were conducted for RUIs, revealing minimal relevance to ICB response (as detailed in lines 174-179 and Supp. Figure 2C-D). These findings have shown that RDIs are significantly enriched for interactions

involved in ICB response compared to RUIs (Fisher's exact test $P = 3.3 \times 10^{-44}$ and 3.2×10^{-40} ; odds ratio = ~18.8 and ~31.1 respectively).

We additionally investigated the association of individual RDIs with survival benefit in terms of progression-free survival and overall survival in the combined set of pre-treatment ICB samples. Our analysis revealed that only a subset of RDIs is pertinent to patient survival, with 74 (25%) and 60 (20%) of RDIs' activity significantly associated with survival benefit using a univariable Cox hazard model for overall and progression-free survival, respectively. Conversely, when the same analysis was applied to RUIs, RDIs were significantly more enriched for interactions implicated in ICB survival compared to RUIs (Fisher's exact test $P = 4.09 \times 10^{-10}$ and 3.98×10^{-6} ; odds ratio = ~4.78 and ~3.28; see Figures).

Overall, our results emphasize the collective high relevance of RDIs to ICB therapy outcome, even though not all are significantly associated. Additionally, considering that biological systems operate via pathways, we demonstrated that an ensemble of these RDIs indicates a better impact on ICB therapy response. We have incorporated these results into Supplementary Table 1, including individual odds ratios and FDR (False Discovery Rate) for each RDI interaction occurring in responders vs. non-responders (pre-treatment and on-treatment separately), as well as hazard ratios and FDR for each interaction in terms of both overall and progression-free survival.

We have incorporated Figure R3C3F1 into the manuscript as supplementary Figures 2E-H and have appended several sentences to expound upon these findings. These additions can be found in lines 179-186 within the Results section of the revised manuscript:

“Moreover, we also studied the association of each individual interaction with both overall and progression-free survival benefit in the combined set of pre-treatments ICB samples. Similarly, the results reveal that RDIs are significantly more enriched for interactions implicated with ICB overall and progression-free survival compared to RUIs (Fisher's exact test $P = 4.09 \times 10^{-10}$ and 3.98×10^{-6} ; odds ratio = ~4.78 and ~3.28; see Supp Fig. 2E-H). These findings together highlight the potential functional importance of RDIs in mediating ICB resistance, in contrast to the RUIs that lack predictive power. Therefore, we focus on RDIs in our subsequent analyses.”

Figure R3C3F1: The association between individual RDI interaction and patient survival. Univariable cox proportional hazard regression analysis identified individual activated RUIs or RDIs in the combined set of pre-treatment samples receiving ICB therapy with overall survival and progression free survival annotations. The X-axis indicates the hazard ratio of individual interactions based on their activity (activated or inactivated) either providing beneficial survival outcomes (< 0) or adverse survival outcomes (> 0). The Y-axis indicates the significance (FDR) of individual interactions hazard ratios. A-B: RUIs with a hazard ratio > 0 and FDR < 0.2 per cell type pair are considered significantly associated with adverse overall and progression-free survival outcome respectively in pre-treatment samples and are highlighted in blue. C-D: RDIs with a hazard ratio > 0 and FDR < 0.2 per cell type pair are considered significantly associated with beneficial overall and progression-free survival outcome respectively in pre-treatment samples and are highlighted in magenta.

4. *Why are they different between cohorts, are there any specific for a specific checkpoint inhibitor?*

Response:

Thanks. First, in response to a previous comment of yours, we have added now an analysis showing that there is a marked significant overlap between RDIs shared across different ICB cohorts. There is naturally some remaining disparity in the inferred RDIs between cohorts, which can be attributed to various factors. These include potential batch effects,

discrepancies in platforms utilized (such as differences in bulk tumor sample preparation, whether FFPE or fresh frozen), clinical diversities stemming from different medical centers and patient populations, and variations in treatment duration, among others. Moreover, treatments administered fall into three distinct categories (anti-PD1, anti-CTLA4, and combination therapy with both anti-PD1 and anti-CTLA4 agents). Ideally, we would analyze treatment-specific ligand-receptor interactions. Regrettably, due to the limited sample size within each treatment category of the training data, achieving this level of granularity proves unfeasible.

This is now briefly related to in the discussion, appearing in lines 483-488:

“Furthermore, treatments administered fall into three distinct categories (anti-PD1, anti-CTLA4, and combination therapy with both anti-PD1 and anti-CTLA4 agents). Regrettably, due to the limited sample size within each treatment category of the training data, we were unable to extract treatment-specific insights. However, prospective studies in the future could delve into this direction provided there is access to more abundant samples.”

5. *The authors mention that the superiority of their prediction method over other is that they can identify potential targets, but how would you target so many interactions - or how would you identify the very relevant ones?*

Response:

Indeed, this is an important challenge. Our method has identified an assembly of approximately 100 ligand-receptor interactions, collectively serving as a robust biomarker of ICB response in patients. While many of these interactions exhibit nearly equal predictive power, with individual predictive power (and hence potential functional effects in isolation) that is quite low, indicating that they probably act in concert, we also observed prominent interactions displaying significant predictive capability (5 individual interactions with AUC exceeding 0.65 in pre-treatment samples). Additionally, we have demonstrated that the activity of these interactions diminishes as resistance to ICB treatment emerges. While our signature comprises numerous interactions, we concur with the reviewer that targeting such a large number of interactions may not be feasible. Future prospective studies may, however, investigate the functionality of these top interactions and identify hotspot receptor targets for potential therapeutic intervention.

This is now discussed accordingly in lines 497-500:

“Lastly, while our RDIs are composed of ligand-receptors situated on the surface of cells, targeting such large number of interactions in concert is infeasible. Hopefully, future

prospective studies could aim to identify a subset of hotspot receptor targets from these interactions for potential therapeutic intervention (23).”

6. *The CXCR4 - CXCL12 interaction has been described in other studies to be involved in preventing CD8 T cell infiltration in tumour, however the authors here say the opposite, could they explain more?*

Response:

We appreciate the insight provided by the reviewer. We conducted a thorough literature review regarding the CXCL12-CXCR4 axis in recruiting CD8+ T cells to the TME. We discovered the discrepancy of the CXCL12-CXCR4 axis was due to the level-dependence of CXCL12 in the axis. Expression of CXCL12 by melanoma acts as chemoattractant to recruit effector CD8+ T cells to the TME. However, high concentration of CXCL12 can also induce chemo repulsive effects in blocking CD8+ T cells to the TME. To resolve this, we added to our current citations with the original paper that observed both CXCL12-CXCR4 dual chemoattractant and chemorepulsive effects (24). The prior cited references Richmond et al. 2009 and Kohli et al. 2022 supported CXCL12-CXCR4 chemoattractant effects while also alluding to it being dependent on CXCL12 expression (25,26).

We also added to the discussion, “Moreover, varying concentrations of chemokine ligands can either mediate or prevent immune infiltration in the TME. For example, CXCL12-CXCR4 axis facilitates recruitment of CD8+ T cells to the TME. However, at high CXCL12 concentrations the effects become inhibitory” in lines 491-494. As well as modified the following sentence in the discussion to include the term “concentrations” in lines 494-497. It reads as follow: “To address this knowledge gap, one potential solution is to computationally assess the association between each interaction and a clinical phenotype of interest across diverse cell types, concentrations, and contexts, which is of course a complex challenge on its own.”

7. *For the scRNA-seq data, the authors don't mention how or whether they reanalysed the data in the Methods. Did they use already annotated cells and they regroup them into global cell types (they mention they reannotated it to match the bulk data, but I could not find any methods for the analysis)?*

Response:

Thanks. The single-cell data utilized in our study was obtained from Jerby-Arnon et al.'s research (10), encompassing eight distinct cell types as per the original annotation.

Recognizing the pivotal role of dendritic cells (DCs) in antigen presentation and immune response modulation, we embarked on reannotating Livnat's single-cell data to incorporate DCs in our previous study (9). Regrettably, due to editorial constraints, we were obliged to condense the content featured in our previous publication, resulting in the omission of the single-cell re-annotation.

Given that monocytes serve as precursors for both macrophages and DCs, we reannotated cells initially labeled as macrophages. Employing Principal Component Analysis (PCA) on 12 classical DC marker genes (refer to the table below) (27,28), we proceeded with K-means clustering based on the top two principal components, resulting in the identification of four clusters. Cluster 1 was annotated as skin resident dendritic cells (a blend of dermal and epidermal DCs), while Cluster 2 was designated as plasmacytoid DCs (pDCs). These two additional cell type labels were integrated into Livnat's original single-cell data for subsequent downstream deconvolution analyses.

Figure R3C7F1: An illustration depicting the clustering of pre-labeled Macrophages based on 12 classical DC marker genes. Following PCA analysis, K-means clustering was utilized to categorize macrophages, revealing four distinct clusters.

To provide further clarity within the manuscript, we have included the following phrases in the method section in lines 585-590:

“To ensure consistency across the bulk datasets and incorporate dendritic cells (DCs), which play a crucial role in immune response through antigen presentation modulation, we reannotated subsets of macrophages as DCs. Utilizing K-means clustering and PCA analysis on twelve classical DC marker genes (27,28), this expanded the number of cell types from eight to ten. The reannotation encompassed skin resident DCs (a blend of dermal and epidermal DCs) along with plasmacytoid DCs.”

DC subset	Classical markers
pDC	IL3RA (CD123) CLEC4C (BDCA2) NRP1 (BDCA4)
cDC1	THBD (BDCA3)
cDC2	CD1C (BDCA1) ITGAX (CD11c) ITGAM (CD11b)
Langerhans	CD207 (Langerin) CD1A CDH1 (E-cadherin)
Non-classical	FCGR3A (CD16) CX3CR1

Table R3C7T1: The classical gene markers for different subset of dendritic cells.

8. *I appreciate the reason why the authors used global cell type clusters in the scRNA-seq data, to show that the method also works there, but they can expand these analysis and utilise the heterogeneity and granularity of cell types/states that scRNA-seq data provides to look more in depth of which more detailed cell types have expression of the predicted ligand-receptor interactions to get a more defined hypothesis of a potential mechanism.*

Response:

We appreciate the reviewer’s insightful suggestion. We acknowledge the potential benefits of extending our analyses to a higher resolution of cell types, which could provide deeper mechanistic insights into ICB therapy. However, it would pose a significant challenge for deconvolution methods such as CODEFACS, as their performance is influenced by the abundance of cell types (9). It is probable that obtaining accurate estimations of cell type-specific expression profiles through deconvolution would be difficult when working with more detailed cell subtypes. To address this concern and respond to the reviewer’s suggestion, we propose conducting cell-cell interaction analysis at a higher resolution of cell types using the single-cell ICB cohort from Jerby-Arnon et al.’s study (10). This approach allows us to assess the relevance of the RDIs identified in our bulk cohorts to specific cell subtypes in the single-cell cohort.

Our focus was specifically on CD4+ and CD8+ T cell states, which have been meticulously annotated by Jerby-Arnon et al. (10) as cytotoxic, exhausted, naïve, and Treg states. Utilizing

the same in-house tool, SOCIAL (described in detail in our response to your comment 1), which we have employed for single-cell transcriptomics, we inferred the activity of ligand-receptor interactions from the single-cell transcriptomics data (as described in the Methods) across these detailed cell states. Regrettably, our analysis did not reveal any cell state in which the RDIs were overrepresented. To elaborate, among the 74 CD8+ T cell relevant interactions identified in our RDI list, only up to 6 interactions were significantly activated in individual cell states (refer to Figure R3C8F1). Similarly, for CD4+ T cells, out of the 49 CD4+ T cell relevant interactions from our RDI list, only up to 8 interactions were significantly activated in individual cell states (refer to Figure R3C8F2). Consequently, we conclude that our RDIs lack specificity to any particular detailed cell subtypes.

Given the absence of specific additional insights from these analyses, we have opted not to include these results in our manuscript.

Figure R3C8F1. CD8+ T cell state contributions to individual ligand-receptor interactions. The X-axis corresponds to a subset of RDIs' (read as *ligand-cell-type_receptor-cell-type_ligand-gene_receptor-gene*) that are significantly overrepresented/activated in at-least one CD8+ T cell state. The Y-axis indicates the CD8+ T cell state and the size of the magenta dots corresponds to the significance (FDR) of each interaction enrichment within each CD8+ cell state (Fisher's one-sided test). The background for each interaction within each CD8+ cell state included the activity of the remaining CD8+ T cell states.

Figure R3C8F2. CD4+ T cell state contributions to individual ligand-receptor interactions. The X-axis corresponds to a subset of RDIs’ (read as *ligand-cell-type_receptor-cell-type_ligand-gene_receptor-gene*) that are significantly overrepresented/activated in at-least one CD4+ T cell state. The Y-axis indicates the CD4+ T cell state and the size of the magenta dots corresponds to the significance (FDR) of each interaction enrichment within each CD4+ cell state (Fisher’s one-sided test). The background for each interaction within each CD4+ cell state included the activity of the remaining CD4+ T cell states.

Additional remarks:

We thank the reviewers for the edits and reviews provided for this manuscript. In addition to implementing the insightful comments provided by the reviewers, we have also implemented additional modifications on our own initiative to further correct and enhance the flow and readability of this manuscript.

Across the entirety of the manuscript, we decided to rename our “resistance activated interactions” or RAIs to “resistance upregulated interactions” or RUIs. Similarly, we renamed our “resistance deactivated interactions” to “resistance downregulated interactions” (still RDIs). Additionally, the “resistance activated score” and “resistance deactivated score” has been changed to “resistance upregulated score” or (RUS) and “resistance downregulated score” (still RDS). We believe this modification will significantly enhance the interpretability of these interactions dynamics as either being upregulated or downregulated following checkpoint therapy to lead to resistance. Moreover, we have relabeled the term “on-treatment” to “post-treatment” to reinforce that these samples have already been exposed to checkpoint therapy and stay consistent with the machine learning method described in now Figure 1. Both corrections have been made across the main-text, figures, and figure legend.

In the results section we modified figure 1 and its legend to include subfigure figure 1A and figure 1C (see Figure 11). Figure 1A describes the pre-processing deconvolution step, using CODEFACS and LIRICS, required to develop a cell-type-specific ligand-receptor interaction profile from bulk transcriptomics. Figure 1C visualizes the demographic of each deconvolved ICB transcriptomics cohort that we studied.

Figure 1: Overview of IRIS. (A-B) The IRIS input includes cell-type-specific ligand-receptor interaction activity profiles (inferred by applying CODEFACS and LIRICS on the tumor transcriptomics) and treatment response labels: responder (R) and non-responder (NR). It consists of two steps: Step I use a Fisher’s test to identify differentially activated ligand-receptor interactions in the pre-treatment and non-responder post-treatment samples. These interactions are categorized as either resistant downregulated interactions (RDI) or resistant upregulated interactions (RUI) based on their differential activity state in the post-treatment vs. the pre-treatment state; that is, RDIs are downregulated in post-treatment resistant patients and vice versa for RUIs. Step II employs a hill climbing aggregative feature selection algorithm to choose the optimal set of RDIs or RUIs for classifying responders and non-responders in pre-treatment samples. The final output of IRIS is a selected set of RDIs and RUIs hypothesized to facilitate in ICB resistance, that can be used to predict ICB therapy response in a new ICB cohort. (C) Demographics of deconvolved melanoma ICB cohorts used to train and validate IRIS. The X-axis indicates the ICB cohort name. The right and left panels correspond to post-treatment and pre-treatment samples to ICB respectively. The top panel depicts the relative proportion of responders (R) and non-responders (NR) samples in each cohort. The middle panel denotes tumor sample size in each cohort. The bottom panel displays the ICB treatment regimen administered, indicated with a purple dot, within each cohort: anti-PD1 monotherapy (aPD1), anti-CTLA4 monotherapy (aCTLA4), and combination therapy (aPD1+aCTLA4).

In addition, within the results section, we corrected the p-value and figure corresponding to figure 2I to 1.81×10^{-37} and the AUC and figure corresponding to figure 2J to 0.87 to maintain

consistency after implementing the more stringent interaction activity inference tool for single-cell transcriptomics that was used for the spatial transcriptomics data sets (see Reviewer 3 Comment 1). Additional minor formatting edits to figure 2C were made to enhance legibility of AUCs.

Figure 2: RDS based prediction of ICB response in melanoma. (A) Boxplot depicting the distribution of AUCs in classifying responder vs. non-responder samples in all melanoma ICB cohorts between inferred resistant downregulated (RDI/RDS) or resistant upregulated interactions (RUI/RUS). **(B)** Boxplot depicting the distribution of resistance downregulated score (RDS) between responder (R) and non-responder (NR) melanoma ICB samples. **(C)** Bar plot depicting the AUC in classifying responder vs. non-responder melanoma samples for numerous published transcriptomic based prediction scores including RDS, IMPRES (1), TIDE (29), MPS (21), Cytotoxic signature (19), and functional ICB resistance (resF) (10). **(D)** Bar plots depicting the odds-ratio in classifying true responder and non-responder samples using RDS

scores. (E) Kaplan-Meier plot depicting overall survival of the combined set of pre-treatment melanoma samples receiving immune checkpoint blockade (N=274). The patients are stratified into low-risk/high-risk groups based on the RDS median value. (F) Kaplan-Meier plot depicting progression-free survival of the combined set of pre-treatment melanoma samples receiving immune checkpoint blockade (N=206). The patients are stratified into low-risk/high-risk groups as above. (G) Kaplan-Meier plot depicting overall survival of the combined set of pre-treatment TCGA-SKCM samples (N=438). The patients are stratified into low-risk/high-risk groups as above. (H) Kaplan-Meier plot depicting progression-free survival of the combined set of pre-treatment TCGA-SKCM samples (N=423). The patients are stratified into low-risk/high-risk groups as above. (I) Boxplot depicting the distribution of RDS between untreated (naïve) and post-checkpoint resistance (post-ICB resistant) samples from a melanoma ICB single-cell cohort. (J) ROC curve depicting the classification accuracy of naïve vs. post-ICB resistant tumors in a melanoma ICB single-cell cohort based on the RDS scores of each patient sample (Methods).

Moreover, in the results section we reordered supplementary figure 4 to supplementary figure 2 and reordered the remaining supplementary figures accordingly to enhance flow. Supplementary figure 4 (now 2) legend was also both substantially simplified and corrected to say two-sided fisher's test, not one-sided test based on the analysis that was performed.

Furthermore, we modified figure 3 in the main text (see Figure 13). In addition to the correction made in Reviewer 3 comment 1, we added a sentence in figure 3A legend describing how to read the functional annotations of each interaction. It reads as follow: “**Literature-curated interaction functions from the LIRICS database are annotated in the inner ring and correspond to the colors detailed in the legend.**” In Figure 3B, we changed the visualization from a boxplot to a lollipop plot and added small notes in the legend guiding readers to distinguish which color corresponds to ligands and which color corresponds to receptors. We also added/implemented figure 3D in the main text, a diagram visualizing a subset of chemokine ligand-receptor interactions found within our RDI network that are associated with lymphocyte infiltration in the tumor microenvironment (TME).

Figure 3: Resistant deactivated interactions are enriched in those known to mediate CD8+ T cell infiltration to the TME. (A) LIRICS' chord diagram representing the 122 interactions (out of 299 interactions in the RDI network) that are overlapped in at least two ICB cohort-specific models. Cell type abbreviations detailed in the methods. Literature-curated interaction functions from the LIRICS database are annotated in the inner ring and correspond to the colors detailed in the legend. (B) Lollipop plot depicting enrichment of individual cell types within RDI network as ligand-expressing (in red) or receptor-expressing (in blue) cells (the background is all LIRICS' tumor-immune interactions; N=3776). Cell types with enrichment p -values < 0.05 (dotted line) are shown. (C) Bar plot depicting enrichment of functional annotations within RDI network (background is the same as above). Functional annotations with p -values < 0.05 (dotted line) are shown. (D) Diagram displaying examples of cell-type-specific chemokine interactions found within the RDI network associated with CD8+ T cell infiltration in the TME supported by literature.

Additionally, in the results section in lines 423-424, we emphasized that the mechanism of ICB resistance that we propose is theoretical. It reads as follows: **“We have proposed a theoretical mechanism underlying the development of resistance to ICB”**. In addition, we reconceptualized Figure 6 (previously figure 5) to further emphasize that the ensemble downregulation of RDIs results in the development of resistance to ICB (see R2C7F1).

In the discussion section, following our spatial validation (see Reviewer 2 comment 5), we omitted the limitation regarding not incorporating spatial proximity information in our analysis.

In the methods section, we corrected the information regarding the source of Auslander et al. samples to fresh frozen from FFPE in line 509, as well as changed the source of Jerby-Arnon et al. samples to resected tumors instead of FFPE in line 583. Aside from the major edits, minor grammatical and formatting edits were made throughout the manuscript.

References:

1. Auslander N, Zhang G, Lee JS, Frederick DT, Miao B, Moll T, et al. Robust prediction of response to immune checkpoint blockade therapy in metastatic melanoma. *Nat Med* [Internet]. 2018;24(10):1545–9. Available from: <https://doi.org/10.1038/s41591-018-0157-9>
2. Gide TN, Quek C, Menzies AM, Tasker AT, Shang P, Holst J, et al. Distinct Immune Cell Populations Define Response to Anti-PD-1 Monotherapy and Anti-PD-1/Anti-CTLA-4 Combined Therapy. *Cancer Cell* [Internet]. 2019 Feb 11;35(2):238-255.e6. Available from: <https://doi.org/10.1016/j.ccell.2019.01.003>
3. Liu D, Schilling B, Liu D, Sucker A, Livingstone E, Jerby-Arnon L, et al. Integrative molecular and clinical modeling of clinical outcomes to PD1 blockade in patients with metastatic melanoma. *Nat Med* [Internet]. 2019;25(12):1916–27. Available from: <https://doi.org/10.1038/s41591-019-0654-5>
4. Riaz N, Havel JJ, Makarov V, Desrichard A, Urba WJ, Sims JS, et al. Tumor and Microenvironment Evolution during Immunotherapy with Nivolumab. *Cell* [Internet]. 2017 Nov 2;171(4):934-949.e16. Available from: <https://doi.org/10.1016/j.cell.2017.09.028>
5. Cui C, Xu C, Yang W, Chi Z, Sheng X, Si L, et al. Ratio of the interferon- γ signature to the immunosuppression signature predicts anti-PD-1 therapy response in melanoma. *NPJ Genom Med* [Internet]. 2021;6(1):7. Available from: <https://doi.org/10.1038/s41525-021-00169-w>
6. Sharma P, Hu-Lieskovan S, Wargo JA, Ribas A. Primary, Adaptive, and Acquired Resistance to Cancer Immunotherapy. *Cell* [Internet]. 2017 Feb 9;168(4):707–23. Available from: <https://doi.org/10.1016/j.cell.2017.01.017>
7. Morad G, Helmink BA, Sharma P, Wargo JA. Hallmarks of response, resistance, and toxicity to immune checkpoint blockade. Vol. 184, *Cell*. Elsevier B.V.; 2021. p. 5309–37.
8. Huang AC, Zappasodi R. A decade of checkpoint blockade immunotherapy in melanoma: understanding the molecular basis for immune sensitivity and resistance. *Nat Immunol* [Internet]. 2022;23(5):660–70. Available from: <https://doi.org/10.1038/s41590-022-01141-1>
9. Wang K, Patkar S, Lee JS, Gertz EM, Robinson W, Schischlik F, et al. Deconvolving Clinically Relevant Cellular Immune Cross-talk from Bulk Gene Expression Using CODEFACS and LIRICS Stratifies Patients with Melanoma to Anti-PD-1 Therapy. *Cancer Discov* [Internet]. 2022 Apr 1;12(4):1088–105. Available from: <https://doi.org/10.1158/2159-8290.CD-21-0887>
10. Jerby-Arnon L, Shah P, Cuoco MS, Rodman C, Su MJ, Melms JC, et al. A Cancer Cell Program Promotes T Cell Exclusion and Resistance to Checkpoint Blockade. *Cell* [Internet]. 2018;175(4):984-997.e24. Available from: <https://www.sciencedirect.com/science/article/pii/S0092867418311784>
11. Kumar MP, Du J, Lagoudas G, Jiao Y, Sawyer A, Drummond DC, et al. Analysis of Single-Cell RNA-Seq Identifies Cell-Cell Communication Associated with Tumor Characteristics. *Cell Rep* [Internet]. 2018;25(6):1458-1468.e4. Available from: <https://www.sciencedirect.com/science/article/pii/S221112471831636X>
12. Vento-Tormo R, Efremova M, Botting RA, Turco MY, Vento-Tormo M, Meyer KB, et al. Single-cell reconstruction of the early maternal–fetal interface in humans. *Nature*

- [Internet]. 2018;563(7731):347–53. Available from: <https://doi.org/10.1038/s41586-018-0698-6>
13. Vahid MR, Brown EL, Steen CB, Zhang W, Jeon HS, Kang M, et al. High-resolution alignment of single-cell and spatial transcriptomes with CytoSPACE. *Nat Biotechnol* [Internet]. 2023;41(11):1543–8. Available from: <https://doi.org/10.1038/s41587-023-01697-9>
 14. Francis K, Palsson BO. Effective intercellular communication distances are determined by the relative time constants for cyto/chemokine secretion and diffusion. *Proceedings of the National Academy of Sciences* [Internet]. 1997 Nov 11;94(23):12258–62. Available from: <https://doi.org/10.1073/pnas.94.23.12258>
 15. Thrane K, Eriksson H, Maaskola J, Hansson J, Lundeberg J. Spatially Resolved Transcriptomics Enables Dissection of Genetic Heterogeneity in Stage III Cutaneous Malignant Melanoma. *Cancer Res* [Internet]. 2018 Oct 15;78(20):5970–9. Available from: <https://doi.org/10.1158/0008-5472.CAN-18-0747>
 16. Biermann J, Melms JC, Amin AD, Wang Y, Caprio LA, Karz A, et al. Dissecting the treatment-naïve ecosystem of human melanoma brain metastasis. *Cell* [Internet]. 2022;185(14):2591–2608.e30. Available from: <https://www.sciencedirect.com/science/article/pii/S0092867422007127>
 17. Stuart T, Butler A, Hoffman P, Hafemeister C, Papalexi E, Mauck WM, et al. Comprehensive Integration of Single-Cell Data. *Cell* [Internet]. 2019;177(7):1888–1902.e21. Available from: <https://www.sciencedirect.com/science/article/pii/S0092867419305598>
 18. Cable DM, Murray E, Zou LS, Goeva A, Macosko EZ, Chen F, et al. Robust decomposition of cell type mixtures in spatial transcriptomics. *Nat Biotechnol* [Internet]. 2022;40(4):517–26. Available from: <https://doi.org/10.1038/s41587-021-00830-w>
 19. Davoli T, Uno H, Wooten EC, Elledge SJ. Tumor aneuploidy correlates with markers of immune evasion and with reduced response to immunotherapy. *Science (1979)* [Internet]. 2017 Jan 20;355(6322):eaaf8399. Available from: <https://doi.org/10.1126/science.aaf8399>
 20. Ayers M, Lunceford J, Nebozhyn M, Murphy E, Loboda A, Kaufman DR, et al. IFN- γ -related mRNA profile predicts clinical response to PD-1 blockade. *J Clin Invest* [Internet]. 2017 Aug 1;127(8):2930–40. Available from: <https://doi.org/10.1172/JCI91190>
 21. Pérez-Guijarro E, Yang HH, Araya RE, El Meskini R, Michael HT, Vodnala SK, et al. Multimodel preclinical platform predicts clinical response of melanoma to immunotherapy. *Nat Med* [Internet]. 2020;26(5):781–91. Available from: <https://doi.org/10.1038/s41591-020-0818-3>
 22. Steiniche T, Rha SY, Chung HC, Georgsen JB, Ladekarl M, Nordmark M, et al. Prognostic significance of T-cell–inflamed gene expression profile and PD-L1 expression in patients with esophageal cancer. *Cancer Med* [Internet]. 2021 Dec 1;10(23):8365–76. Available from: <https://doi.org/10.1002/cam4.4333>
 23. Armingol E, Officer A, Harismendy O, Lewis NE. Deciphering cell–cell interactions and communication from gene expression. *Nat Rev Genet* [Internet]. 2021;22(2):71–88. Available from: <https://doi.org/10.1038/s41576-020-00292-x>
 24. Zhang T, Somasundaram R, Berking C, Caputo L, Van Belle P, Elder DE, et al. Preferential involvement of CX chemokine receptor 4 and CX chemokine ligand 12 in T-Cell migration toward melanoma cells. *Cancer Biol Ther* [Internet]. 2006 Oct 11;5(10):1304–12. Available from: <https://doi.org/10.4161/cbt.5.10.3153>

25. Kohli K, Pillarisetty VG, Kim TS. Key chemokines direct migration of immune cells in solid tumors. *Cancer Gene Ther* [Internet]. 2022;29(1):10–21. Available from: <https://doi.org/10.1038/s41417-021-00303-x>
26. Richmond A, Yang J, Su Y. The good and the bad of chemokines/chemokine receptors in melanoma. *Pigment Cell Melanoma Res* [Internet]. 2009 Apr 1;22(2):175–86. Available from: <https://doi.org/10.1111/j.1755-148X.2009.00554.x>
27. Merad M, Sathe P, Helft J, Miller J, Mortha A. The Dendritic Cell Lineage: Ontogeny and Function of Dendritic Cells and Their Subsets in the Steady State and the Inflamed Setting. *Annu Rev Immunol* [Internet]. 2013;31(Volume 31, 2013):563–604. Available from: <https://www.annualreviews.org/content/journals/10.1146/annurev-immunol-020711-074950>
28. Collin M, Bigley V. Human dendritic cell subsets: an update. *Immunology* [Internet]. 2018 May 1;154(1):3–20. Available from: <https://doi.org/10.1111/imm.12888>
29. Jiang P, Gu S, Pan D, Fu J, Sahu A, Hu X, et al. Signatures of T cell dysfunction and exclusion predict cancer immunotherapy response. *Nat Med* [Internet]. 2018;24(10):1550–8. Available from: <https://doi.org/10.1038/s41591-018-0136-1>

REVIEWERS' COMMENTS

Reviewer #1 (Remarks to the Author):

Authors have provided a revised version of the manuscript which addressed most of the previous concerns. The changes made have significantly enhanced the clarity and quality of the paper. I have no further major comments

Reviewer #2 (Remarks to the Author):

The authors are commended for such thorough and extensive revisions and they addressed essentially all of my comments. I would not dispute the notion that the model is in fact novel in its construction and operation. However, the significance of the result leaves something to be desired. The performance is quite variable and many of the comparisons with other, often simpler, methods is either comparable or performs worse (Figure 2C and p-values lines 191-195). The method only significantly outperformed one other method in MPS. Additionally, all results seem to lead to CD8 T cell infiltration and hot vs. cold tumors, as being associated with the response associated signature (RDS). It is widely known and has been well reported that T cell infiltration and hot tumors are predictive of response to ICB, and therefore we're left with no new information about what factors could predict response to ICB. There was an opportunity lost here where the authors could have expanded more on what new ligand-receptor interactions were contributing most to T cell infiltration, which would provide new insight. I am well aware that the model relies on pre-defined ligand-receptor interactions, but some of them may be new in the context of T cell trafficking into tumors or T cell exclusion from tumors.

Reviewer #3 (Remarks to the Author):

The authors have addressed my comments and have significantly improved the manuscript with the addition of spatial and single-cell RNA-seq data.

Reviewer #3 (Remarks on code availability):

The code was not provided.

Responses to reviewers' comments

General comments: Since Reviewers 1 and 3 have no additional comments, we will address the feedback provided by Reviewer 2 in this response letter. The reviewer comments are copied verbatim in black font and alternate with our responses, which are presented in **red bold font**. Any text additions made to the manuscript to address the reviewers' comments are highlighted in **turquoise** and are indented.

Reviewer #2:

The authors are commended for such thorough and extensive revisions and they addressed essentially all of my comments. I would not dispute the notion that the model is in fact novel in its construction and operation. However, the significance of the result leaves something to be desired. The performance is quite variable and many of the comparisons with other, often simpler, methods is either comparable or performs worse (Figure 2C and p-values lines 191-195). The method only significantly outperformed one other method in MPS. Additionally, all results seem to lead to CD8 T cell infiltration and hot vs. cold tumors, as being associated with the response associated signature (RDS). It is widely known and has been well reported that T cell infiltration and hot tumors are predictive of response to ICB, and therefore we're left with no new information about what factors could predict response to ICB. There was an opportunity lost here where the authors could have expanded more on what new ligand-receptor interactions were contributing most to T cell infiltration, which would provide new insight. I am well aware that the model relies on pre-defined ligand-receptor interactions, but some of them may be new in the context of T cell trafficking into tumors or T cell exclusion from tumors.

Response:

Thank you for acknowledging our revision efforts and for your insightful comments.

Firstly, we appreciate the recognition of the novelty in our methodologies. Our study introduces tools and demonstrates how to analyze cell-type-specific ligand-receptor interactions across three different data modalities. Regarding the comment on ICB response prediction, our method achieves an average AUC of 0.72 across datasets, with performance comparable to the cytotoxic signature and superior to four other methods (Fig. 2C). To assess robustness, we have now calculated the Coefficient of Variation ($CV = \frac{sd(AUCs)}{mean(AUCs)} * 100$), given the performance (AUC values) of each method across the datasets. The CV values for our method are markedly smaller than that of the other methods (IMPRES, TIDE, MPS, Cytotoxic signature and resF) 13 vs 25, 23.4, 23.9, 21, 30.1 respectively. Such smaller CV values indicate better robustness, testifying that our method demonstrates greater consistency across datasets.

Accordingly, we have included the following sentence in the Results section, line 179-183 of the manuscript: **“To assess robustness, we calculated the Coefficient of Variation ($CV = \frac{sd(AUCs)}{mean(AUCs)} * 100$), based on the performance (AUC values) of each method across datasets. Our method's CV is substantially smaller ($CV = 13.0$) compared to others ($CV = 24.7$ for IMPRES; 23.3 for TIDE, 24.4 for MPS; 20.9 for Cytotoxic Signature; and 30.3 for resF), demonstrating greater consistency.”**

Although the one-sided Wilcoxon test p-values in Figure 2C are occasionally not significant due to the limited number of datasets (only 8), the overall trends observed indicate the superiority of our method over the other methods. More importantly, our approach provides valuable insights into mechanisms and complements existing methods.

We acknowledge that T cell infiltration and hot tumors are known predictors of ICB response. However, notably, our major findings go beyond that and include: 1) Identification of an optimal subset of ligand-receptor interactions predictive of ICB response, with our functional analysis highlighting their relevance to chemotaxis interactions. 2) Discovery of a fundamentally interesting dynamic process, showing that tumors undergo large-scale negative selection of these chemotaxis-relevant interactions to drive resistance. 3) Systematic assessment of the association between these interactions and T cell infiltration across bulk, single-cell, and spatial transcriptomics data.

We agree that extending our study to explore de novo ligand-receptor interactions related to T cell infiltration or hot TME would be valuable. This falls outside the current scope of our study, we plan to address it in future research. Thank you again for the valuable suggestion.